# Dynamic epistasis analysis reveals how chromatin remodeling regulates transcriptional bursting

Ineke Brouwer[1], Emma Kerklingh[1], Fred van Leeuwen [2,3] & Tineke L. Lenstra [1] ✉

Transcriptional bursting has been linked to the stochastic positioning of nucleosomes. However, how bursting is regulated by the remodeling of promoter nucleosomes is unknown. Here, we use single-molecule live-cell imaging of *GAL10* transcription in *Saccharomyces cerevisiae* to measure how bursting changes upon combined perturbations of chromatin remodelers, the transcription factor Gal4 and preinitiation complex components. Using dynamic epistasis analysis, we reveal how the remodeling of different nucleosomes regulates transcriptional bursting parameters. At the nucleosome covering the Gal4 binding sites, RSC and Gal4 binding synergistically facilitate each burst. Conversely, nucleosome remodeling at the TATA box controls only the first burst upon galactose induction. At canonical TATA boxes, the nucleosomes are displaced by TBP binding to allow for transcription activation even in the absence of remodelers. Overall, our results reveal how promoter nucleosome remodeling together with Gal4 and preinitiation complex binding regulates transcriptional bursting.

The transcription of many genes occurs in stochastic bursts of transcriptional activity, interspersed by periods with no transcriptional activity. To achieve the correct transcriptional output of a bursting gene, the regulatory factors may modulate when a burst starts (burst frequency), when it ends (burst duration) and the rate of polymerase loading during a burst (initiation rate)[1–7]. Yet, it remains elusive how each of these steps is controlled. Previous studies from budding yeast suggested a link between the bursting and chromatin structure[4,8–12]. Inference of transcriptional parameters from mRNA or protein distributions suggests that bursting is affected by mutations in chromatin regulators[8,13,14]. In addition, single-cell mapping of nucleosome conformations at the *PHO5* promoter suggests stochastic transitions between different promoter configurations, of which only some may be permissive for transcription[12,15]. However, how different promoter nucleosome configurations affect the dynamics of transcription in living cells is unexplored.

The positioning of nucleosomes throughout the genome is controlled by chromatin remodeling enzymes. For the promoter regions, the most important remodelers are RSC and SWI/SNF, which together maintain a promoter architecture consisting of a nucleosome-depleted region (NDR) flanked by two nucleosomes referred to as the +1 and −1 nucleosome[16–18]. Upon depletion of RSC, the +1 nucleosome shifts into the NDR for 70% of all genes[19,20]. At highly expressed genes, SWI/SNF acts redundantly with RSC to maintain the NDR[17,18]. In addition, RSC also regulates partially unwrapped unstable nucleosomes, referred to as fragile nucleosomes, that are often found in the promoters of highly expressed genes, such as *GAL10* (refs. 21–25). The interplay between different remodelers at the same promoter is a dynamic process. Individual remodelers remain chromatin bound for just a few seconds and multiple remodelers can sequentially occupy the same promoter within minutes[26].

Nucleosomes affect different steps in the transcription activation process. For example, nucleosomes inhibit transcription factor

[1]Division of Gene Regulation, the Netherlands Cancer Institute, Oncode Institute, Amsterdam, the Netherlands. [2]Division of Gene Regulation, the Netherlands Cancer Institute, Amsterdam, the Netherlands. [3]Department of Medical Biology, Amsterdam UMC, University of Amsterdam, Amsterdam, the Netherlands. ✉e-mail: t.lenstra@nki.nl

(TF) binding and change the residence time of TFs on nucleosomal DNA[4,27–30]. TF binding to nucleosomal DNA is facilitated by remodelers, as a loss of RSC reduces TF occupancy and binding frequency, and increases TF residence time[8,22]. Additionally, movement of the +1 nucleosome in RSC-depleted cells increases nucleosome coverage of the TATA element (TATA) and transcription start site (TSS), which affects TBP binding, preinitiation complex (PIC) assembly and transcription[19,20,31,32]. However, how each of these mechanisms influences transcriptional bursting is unclear.

In this study, we dissected the role of promoter nucleosome remodeling in regulating transcriptional bursting. We acutely depleted RSC and SWI/SNF and used single-molecule live-cell imaging to measure the changes in transcription dynamics at the GAL10 gene in budding yeast. To decipher the regulatory mechanisms of remodeling at specific promoter nucleosomes, we combined remodeler depletion with perturbations of the TF Gal4, PIC components and histones, and analyzed the effect of these single and combined perturbations on each dynamic parameter of transcription using dynamic epistasis analysis. We found that the fragile nucleosome at the Gal4 binding sites is controlled by the redundant action of RSC and Gal4 binding to facilitate consecutive bursts of transcription. The nucleosomes around the TATA and TSS are remodeled in a partially redundant manner by RSC and SWI/SNF to allow for the first burst of transcription after activation. In addition, our results revealed that TBP competes with nucleosomes at the TATA to enable transcription in the absence of chromatin remodelers. Overall, our study exposed how remodeling at different promoter nucleosomes controls the accessibility of the DNA for binding of TFs and the PIC, and how this affects different kinetic parameters of transcriptional bursting.

## Results

### Promoter nucleosome remodeling by RSC affects each burst

Upon addition of the sugar galactose to yeast cells, nucleosomes in the promoter region of GAL10 are remodeled to activate transcription[33]. Micrococcal nuclease digestion with deep sequencing (MNase-seq) with high and low MNase concentrations showed the coverage of stable and fragile nucleosomes, respectively, in inactive and active conditions[23] (Extended Data Fig. 1a–c). As reported previously[4,22,33], in transcriptionally inactive conditions (raffinose), the GAL10 promoter region showed three nucleosomes: a fragile nucleosome at the Gal4 upstream activation sequences (UASs) and two stable nucleosomes at the TATA and TSS. Upon activation with galactose, the TATA nucleosome was evicted, and the TSS nucleosome was moved downstream, creating an NDR. Consistent with previous findings at wide NDRs, the fragile nucleosome at the UASs remained present[21–23].

To dissect how remodeling at each of these three nucleosomes regulates transcriptional bursting, we conditionally depleted the chromatin remodelers from the nucleus, mapped the effect on promoter nucleosome positioning and linked this to the effect on transcription dynamics. First, we depleted RSC, an important nucleosome remodeler controlling both the stable and fragile nucleosomes in promoter regions[19,20,22]. To map the changes in nucleosome positions, we performed MNase-seq in a galactose-rich media in cells where Sth1, the essential catalytic subunit of RSC, was depleted from the nucleus for 60 min using anchor-away[34]. Sth1 depletion was confirmed by a lack of growth on the rapamycin-containing plates (Extended Data Fig. 2a) and imaging of the mScarlet-anchor-away-tagged Sth1 (Extended Data Fig. 2b). In line with previous studies, the RSC depletion led to a fill-in and shortening of NDRs genome wide (Extended Data Fig. 1d,e)[17,19]. We also observed a lower coverage of the fragile nucleosomes in the promoter regions (Extended Data Fig. 1f,g), in agreement with the fragile nucleosomes representing RSC-bound, partially unwrapped nucleosomal intermediates[22]. At the GAL10 locus, RSC predominantly regulated fragile nucleosomes, with a lower coverage over the Gal4 UASs and TSS (Fig. 1a). In addition, a small shift in the TSS nucleosome was observed (Fig. 1b).

To understand how this perturbed chromatin structure upon RSC depletion affects GAL10 transcription dynamics, we used the PP7 technology to directly monitor the transcription in individual cells in real-time (Fig. 1c and Supplementary Video 1). In short, 14 PP7 repeats were introduced endogenously in the 5′UTR of GAL10 and, upon transcription, the PP7 RNA stemloops were bound by PP7-coat protein fused to GFP-Envy[35]. Using widefield fluorescence microscopy, the accumulation of RNAs at the transcription site (TS) was visualized as a bright spot in the nucleus (Fig. 1d and Supplementary Video 1), of which the intensity was tracked over time (Fig. 1e,f). From these intensity traces, the parameters of transcriptional bursting were determined: the active fraction (the fraction of cells that shows a TS within 1 h after galactose addition), the induction time (the time between galactose addition and the first burst), the burst duration, the time between bursts (as a measure for burst frequency) and the burst intensity. The burst size, defined as the total number of RNAs produced during a burst, is dependent on the burst duration and the burst intensity. Upon RSC depletion, we observed an increased induction time (Fig. 1g) and time between bursts (Fig. 1h), whereas the active fraction, burst duration and burst intensity showed minor or no changes (Extended Data Fig. 3a,h–j). Using single-molecule fluorescence in situ hybridization (smFISH), we validated that the effect of RSC depletion on steady-state transcription did not vary between the different cell-cycle stages (Extended Data Fig. 2c). The RSC remodeling of the fragile GAL10 promoter nucleosomes at the UASs and the TSS is thus correlated with changes in the induction time and the start of each burst.

### SWI/SNF remodeling at the TATA regulates the induction time

Promoter nucleosomes at highly expressed genes are remodeled redundantly by RSC and SWI/SNF[17,18]. Therefore, we addressed how SWI/SNF remodeling affects the nucleosome positioning and transcription dynamics at GAL10 by nuclear depletion of Swi2, the catalytic subunit of SWI/SNF (Extended Data Fig. 2a,b). Consistent with previous reports, MNase-seq upon SWI/SNF depletion showed no changes in the genome-wide promoter nucleosome architecture (Extended Data Fig. 1h–k). In contrast, for GAL10, the coverage of stable nucleosomes within the NDR and specifically at the TATA was increased (Fig. 2a). No change in GAL10 fragile nucleosomes was observed (Extended Data Fig. 3k,l). At the level of transcription, depletion of SWI/SNF resulted in an increased induction time (Fig. 2d) but had a minor effect on other GAL10 transcriptional parameters (Fig. 2g and Extended Data Figs. 2c and 3b–e,h–j). The higher nucleosome coverage within the NDR and at the TATA upon SWI/SNF depletion thus affects induction time, but once the cells are activated, this increased coverage does not affect transcription.

### RSC and SWI/SNF remodeling synergically affect transcription

Because of their redundancy[17,18], the effect of simultaneous RSC and SWI/SNF depletion on both the chromatin structure and transcription dynamics is expected to be larger than the combined effect of their individual depletions. To test this, we simultaneously depleted both RSC and SWI/SNF (Extended Data Fig. 2a,b). As expected, a larger effect on stable nucleosomes was observed than for either single depletion, both genome wide (Extended Data Fig. 1l,m) and at the GAL10 locus (Fig. 2b,c), where the TSS nucleosome moved further into the NDR than for either single depletion. However, the coverage at the GAL10 TATA was similar to the coverage in the single SWI/SNF depletion, and the effect on fragile nucleosomes mimicked the effect of single RSC depletion (Extended Data Fig. 1n,o). RSC and SWI/SNF are thus partially redundant and have different functions to remodel the promoter nucleosomes.

To interpret the effect of RSC and SWI/SNF double depletion, we performed epistasis analysis on each dynamic parameter of transcription and called this method dynamic epistasis analysis. Dynamic epistasis analysis is based on classical epistasis analysis, a method to interpret phenotypic growth defects from genetic interactions that

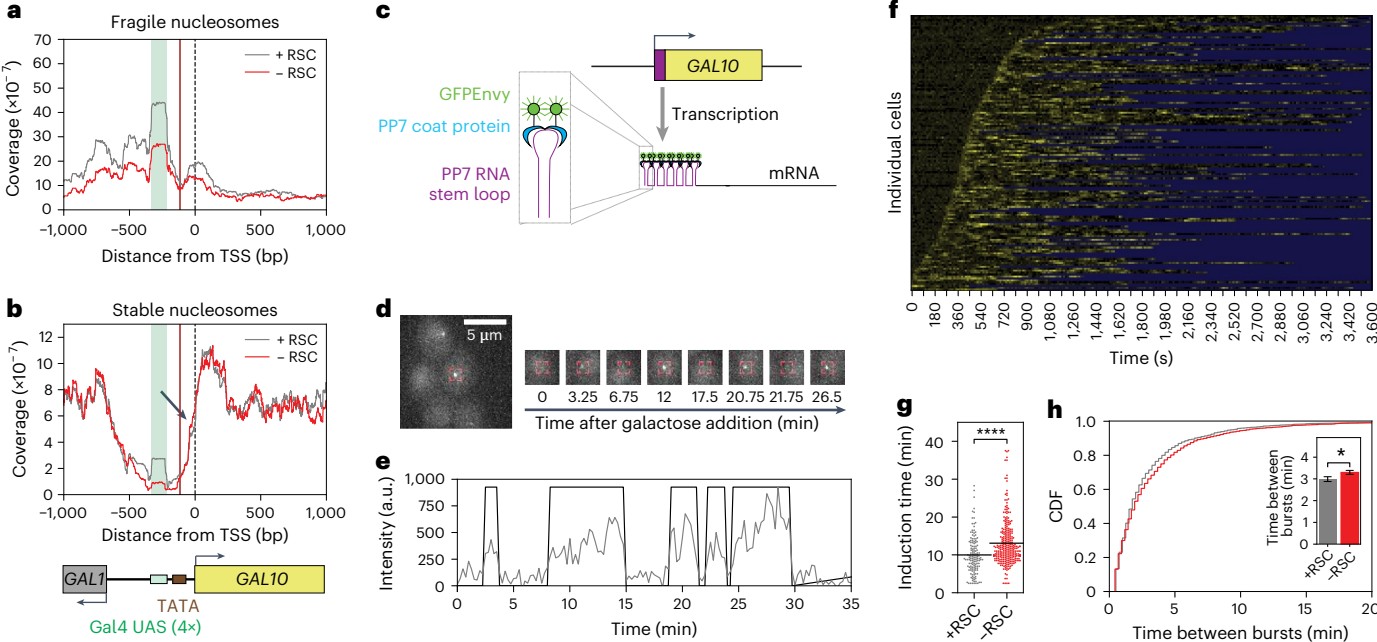

**Fig. 1 | Remodeling of *GAL10* promoter nucleosomes at the UASs and the TSS by RSC affects induction time and time between bursts. a**, MNase-seq analysis of fragile nucleosomes in the *GAL10* promoter region showed a reduced coverage of fragile nucleosomes upon RSC depletion using anchor-away of the catalytic subunit Sth1 by 60 min rapamycin treatment. **b**, MNase-seq analysis of stable nucleosomes in the *GAL10* promoter region. The black arrow indicates a small shift of the +1 nucleosome into the NDR upon RSC depletion. MNase plots in **a** and **b** show one representative replicate out of two experiments. **c**, Schematic of PP7 RNA labeling to visualize *GAL10* transcription in real-time. **d**, Fluorescence signal of an individual TS in a representative cell over time. Red box: location of *GAL10* TS. Representative example out of 143 cells.

**e**, Quantification of TS intensity over time of the cell in (**d**) (gray) and binarization (black). **f**, Heatmap of the TS intensity in $n = 143$ cells (rows) in the presence of RSC. Yellow: fluorescence intensity of TS; blue: region excluded from analysis. **g**, Cells that were depleted of RSC (red) showed an increased *GAL10* induction time compared to the nondepleted cells (gray). $P = <10^{-15}$. **h**, Cells that were depleted of RSC (red) showed an increased time between the burst of *GAL10* transcription compared to the nondepleted cells (gray). Data are presented as cumulative distribution. Inset: data presented as mean values ± s.d. based on 1,000 bootstrap repeats. $P = 0.028$. The significance in **g** and **h** (inset) was determined by two-sided bootstrap hypothesis testing[53]; *, $P < 0.05$, ****, $P < 0.00005$.

allows one to determine whether perturbed factors act in the same or different pathways[36]. Dynamic epistasis analysis compares, for each dynamic parameter of transcription, the observed effect of a double perturbation to the expected effect (the product of the fractional changes in this parameter observed in the individual perturbations). If two perturbations act independently, the double perturbation follows the expected effect (Fig. 2e). Observing a larger effect than expected indicates that the perturbed factors act on the same process and are (partially) redundant, whereas a smaller effect than expected indicates that the factors act in the same pathway or have opposing biochemical functions[36]. For RSC and SWI/SNF double depletion, a larger effect than expected was observed for *GAL10* induction time and time between bursts (Fig. 2d,f,g). In addition, the burst duration and intensity was shorter than expected (Extended Data Fig. 3h–j). As the redundancy between RSC and SWI/SNF was previously established[17,18], the observed synergy for RSC and SWI/SNF depletion validates our approach to identify functional epistatic relationships using transcription dynamics. Moreover, we observed a variation between strains that should theoretically behave the same, such as longer induction times for SWI/SNF and RSC and SWI/SNF than for RSC in nondepleted conditions (Fig. 2d). Such variation could be caused by the anchor-away tags affecting protein function, experiment-to-experiment variation, or additional off-target mutations, although the effects of the latter source were minimized by including 2–3 independent biological replicates. Importantly, dynamic epistasis analysis circumvented this variation and allowed extraction of the specific effects of the perturbations. Overall, these results showed that remodeling of the fragile nucleosomes at the UASs by RSC and nucleosome displacement around the

TATA and the TSS by RSC and SWI/SNF are associated with synergistic changes in the induction time, time between bursts and burst size.

## Remodeling regulates multiple gene-activation steps

To understand how promoter nucleosome remodeling affects the kinetics of transcription activation, we estimated the number of rate-limiting steps from the induction time distributions from the shape parameter $k$ of a Gamma fit. In the presence of remodelers, we found ~6 rate-limiting steps during activation (Extended Data Fig. 4h,n,t). Single or double depletion of RSC and SWI/SNF did not lead to a consistent change in $k$ (Extended Data Fig. 4i,o,u), suggesting that the assumption that each step has an equal rate (underlying the Gamma distribution) is no longer valid. In addition, the high number of steps suggested that multiple rate-liming steps in the signaling pathway dominate and obscure any remodeler-specific steps of transcription activation. To expose remodeling-dependent activation steps, we eliminated signaling steps by pre-exposure with galactose, repression with glucose-containing media, and subsequent re-induction with galactose (Extended Data Fig. 4a). Consistent with previous reports, this re-induction is faster than the initial induction (Extended Data Fig. 4b,c,j,p,v), due to transcriptional memory[33,37]. The re-induction times showed a single rate-limiting step, even when RSC or SWI/SNF was depleted (Extended Data Fig. 4j,k,p,q,v). However, the simultaneous depletion of RSC and SWI/SNF increased the number of steps to ~4 or ~2 depending on the repression conditions (Extended Data Fig. 4w,y), suggesting that RSC and SWI/SNF regulate up to three remodeler-dependent activation steps.

Moreover, similar to the first induction, remodeler depletion during a second galactose exposure resulted in increased induction time

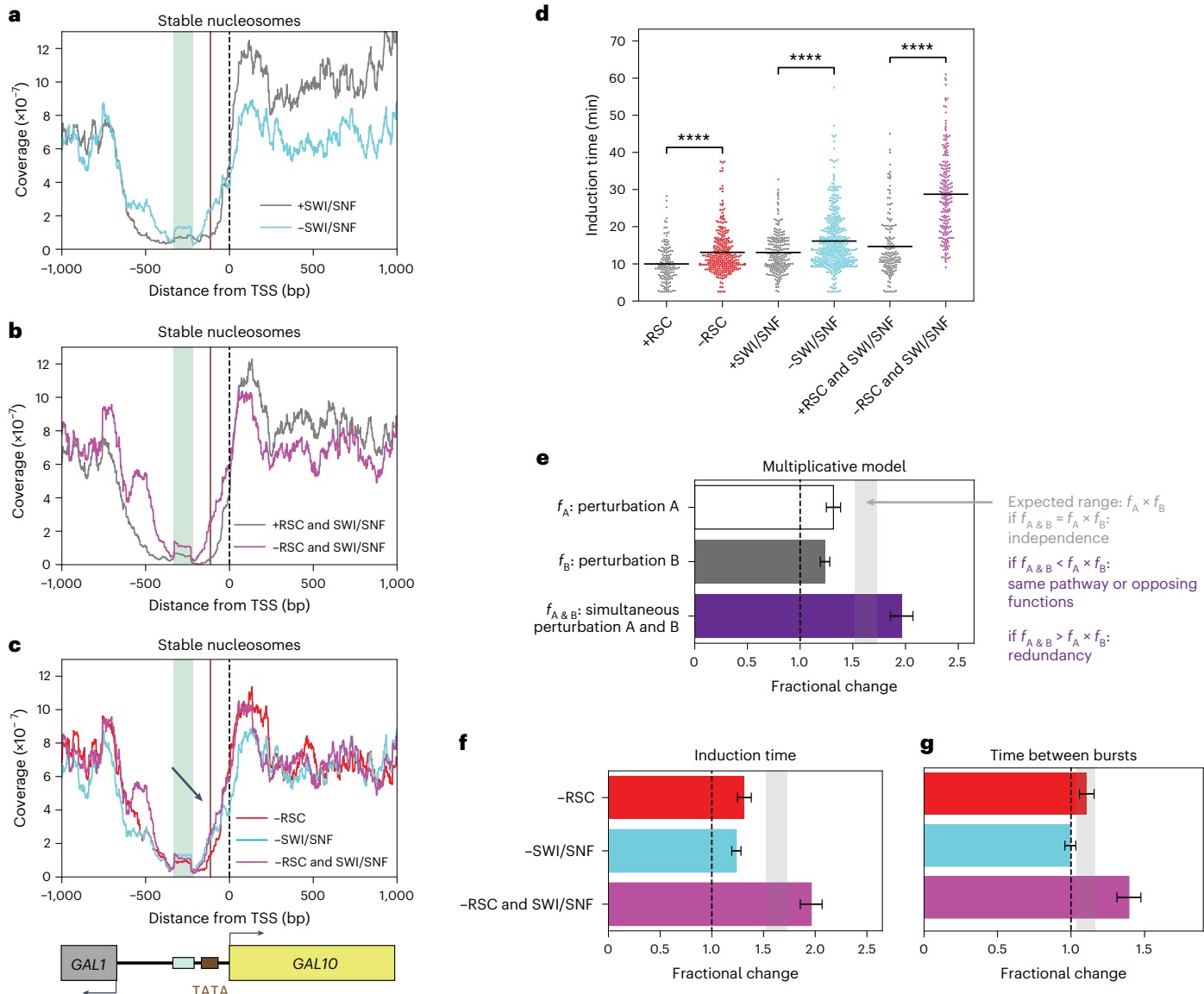

**Fig. 2 | Partially redundant remodeling of nucleosomes by RSC and SWI/SNF at the TATA and TSS synergistically affects the induction time and time between bursts. a**, MNase-seq analysis in the *GAL10* promoter region showed higher coverage of stable nucleosomes around the TATA upon depletion of SWI/SNF by anchor-away of the catalytic subunit Swi2. **b**, MNase-seq analysis of stable nucleosomes in the *GAL10* promoter region showed higher coverage at the TATA and around the TSS upon simultaneous depletion of RSC and SWI/SNF than depletion of either RSC or SWI/SNF individually. **c**, Overlay of MNase-seq analysis of stable nucleosomes in the *GAL10* promoter region upon depletion of RSC, SWI/SNF and simultaneous depletion of RSC and SWI/SNF showed increased coverage (black arrow). MNase plots in **a**–**c** show one representative replicate out of two experiments. **d**, Cells that were depleted of RSC and/or SWI/SNF (red, cyan, magenta) had an increased *GAL10* induction time compared to the nondepleted cells (gray). Significance determined by two-sided bootstrap hypothesis testing[53]; \*\*\*\*, $P < 0.00005$. $P$ values: RSC, $P < 10^{-15}$; ±SWI/SNF, $P < 10^{-15}$; ±RSC and

SWI/SNF, $P < 10^{-15}$. **e**, The multiplicative model for dynamic epistasis analysis was used to assess the effect of double perturbations. The expected effect of a double perturbation for independent processes (grey shaded area) is the product of the effect of the individual perturbations ($f_A$ and $f_B$). If the observed effect ($f_{A\&B}$) is smaller than this expected effect, the perturbations are in the same pathway or have opposing functions. If the observed effect is larger, the processes are redundant. **f**, The increase in *GAL10* induction time when simultaneously depleting RSC and SWI/SNF was larger than expected, based on their individual depletions. Gray bar, expected effect based on dynamic epistasis analysis. **g**, The increase in time between *GAL10* transcriptional bursts in the RSC and SWI/SNF double depletion was larger than expected based on their individual depletions. Gray bar, expected effect based on dynamic epistasis analysis. Data in **e**, **f** and **g** are presented as the fractional change based on mean values ± s.d. based on 1,000 bootstrap repeats.

and time between bursts (Extended Data Fig. 4f). However, the lower burst duration and intensity observed in RSC and SWI/SNF depletion during the first induction was not reproduced during re-induction, suggesting that this effect may be context specific (Extended Data Fig. 4d,e,g). For subsequent analysis, we focused on the induction time and time between bursts. Overall, these memory experiments showed that the remodeling of promoter nucleosomes by RSC and SWI/SNF

promotes multiple activation steps, perhaps by acting on the different promoter nucleosomes.

## Remodeler-free transcription activation

The above experiments showed that simultaneous RSC and SWI/SNF depletion reduces but does not abolish *GAL10* transcription (Fig. 2d,f,g and Extended Data Fig. 3h–j). Also, comparison of the *GAL10* promoter

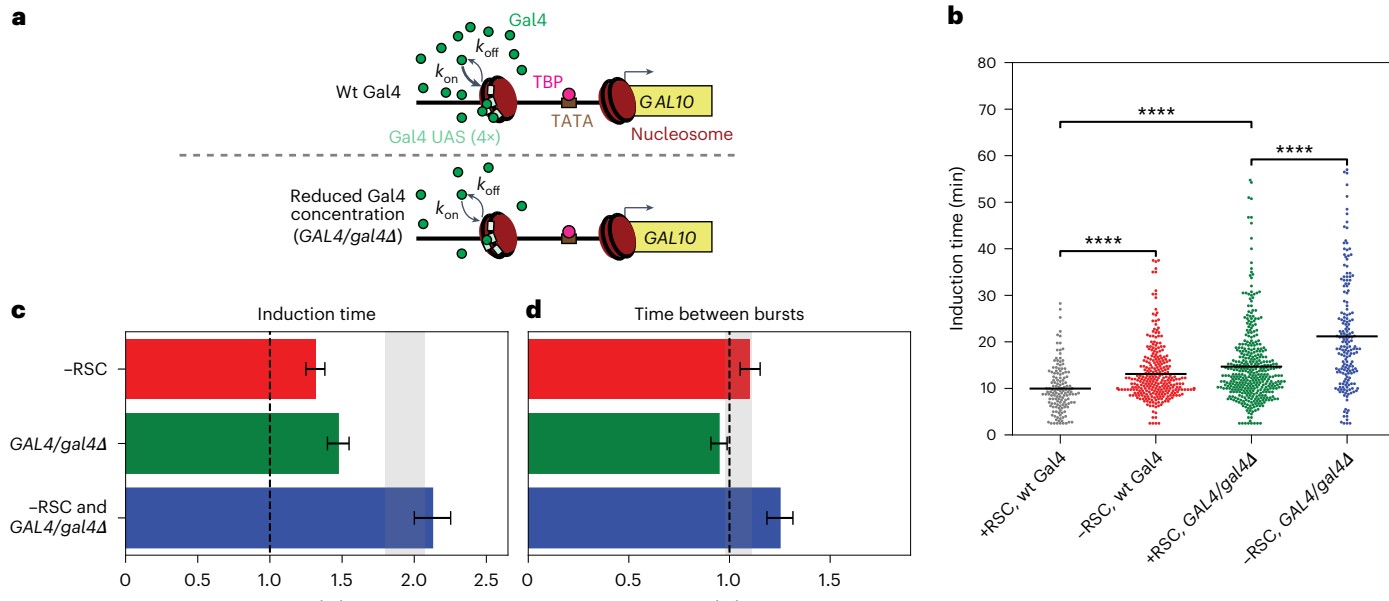

**Fig. 3 | Remodeling of the fragile nucleosome at the UASs by RSC and Gal4 binding synergistically regulates the time between bursts. a**, Schematic showing reduced Gal4 on-rate in a *GAL4/gal4Δ* strain where Gal4 protein concentration was reduced. **b**, *GAL10* induction time was increased in *GAL4/gal4Δ* cells and in RSC-depleted cells. Significance determined by two-sided bootstrap hypothesis testing[53]; ****, $P < 0.00005$. $P$ values: ±RSC, *GAL4/GAL4*, $P < 10^{-15}$; ±RSC, *GAL4/gal4Δ*, $P < 10^{-15}$; +RSC, *GAL4/GAL4* versus +RSC, *GAL4/gal4Δ*,

$P < 10^{-15}$. **c**, Increase in *GAL10* induction time in RSC-depleted *GAL4/gal4Δ* cells was as expected, based on individual perturbations. Gray bar, expected effect based on dynamic epistasis analysis. **d**, The time between consecutive bursts of *GAL10* transcription in *GAL4/gal4Δ* cells increased more than expected based on individual perturbations. Gray bar, expected effect based on dynamic epistasis analysis. Data in **c** and **d** are presented as the fractional change based on mean values ± s.d. based on 1,000 bootstrap repeats.

nucleosome coverage upon RSC and SWI/SNF depletion (Fig. 2b,c) to inactive conditions (Extended Data Fig. 1a–c) indicated that even in the absence of both RSC and SWI/SNF, nucleosomes were remodeled upon galactose induction. This remodeling must occur independently of RSC and SWI/SNF and could either be performed by other remodelers or by the transcriptional machinery itself. To test whether other remodelers are involved, we performed smFISH to detect changes in *GAL10* nascent transcription upon single anchor-away depletion of the catalytic subunit of each of the seven budding yeast remodeling complexes (Extended Data Fig. 3q,r). Apart from RSC, none of the remodeling complexes had a large effect on steady-state transcription levels of *GAL10*. If another remodeler regulates nucleosomes in the *GAL10* promoter, it must act redundantly with RSC and SWI/SNF. Alternatively, the transcriptional machinery itself may perturb nucleosome architecture.

**Remodeling at the Gal4 UASs controls each burst**
The depletion experiments above link the nucleosome changes to changes in transcription dynamics, but both RSC and SWI/SNF depletion affected multiple nucleosomes. To decipher the mechanism by which remodeling at individual nucleosomes controls transcriptional bursting, we combined remodeler depletions with perturbations that affect the binding of regulatory factors at specific nucleosomes. First, we focused on the fragile Gal4 UAS nucleosome. This nucleosome appears to be solely remodeled by RSC, as the coverage at the Gal4 UASs changed upon RSC depletion but not upon SWI/SNF depletion (Fig. 1a,b, Fig. 2a–c and Extended Data Fig. 3k,l). The effect of remodeling of this nucleosome on transcription dynamics was assessed in *GAL4/gal4Δ* cells, where Gal4 expression is reduced twofold (Extended Data Fig. 3m) and where we expect a reduced Gal4 on-rate for UAS binding (Fig. 3a). In these *GAL4/gal4Δ* cells, an increase in induction time was observed (Fig. 3b,c) with a modest effect on other transcriptional parameters (Fig. 3d and Extended Data Fig. 3f,g,n–p). Upon additional depletion of RSC in *GAL4/gal4Δ* cells, the induction time

was within the expected range, suggesting independent roles of Gal4 and RSC during the first transcriptional burst (Fig. 3c). In contrast, we observed a synergistic increase in the time between bursts (Fig. 3d), which indicates that RSC remodeling of the Gal4 UAS nucleosome is redundant with Gal4 binding. Gal4 binding can thus substitute for RSC, possibly by partially unwrapping the nucleosome at the Gal4 UASs. Furthermore, this combined action of RSC remodeling and Gal4 binding at the fragile nucleosome needs to occur before the start of each burst of transcription.

**Synergy of nucleosome remodeling and TBP binding at the TATA**
Next, we focused on the nucleosomes around the *GAL10* TATA and TSS, a region crucial for PIC assembly. One of the first steps in PIC assembly is TBP binding to the TATA, which recruits the rest of the transcription machinery to start transcribing the gene. The TATA is covered by a nucleosome in inactive conditions and is exposed upon activation (Extended Data Fig. 1a–c). The eviction of this nucleosome was reduced in SWI/SNF-depleted cells. In addition, the movement of the TSS nucleosome into the NDR after RSC depletion affects TBP binding to the TATA[19]. To uncover the mechanisms by which remodeling of these nucleosomes affects transcription dynamics, we combined a remodeler depletion with a partial TBP depletion by introducing the anchor-away tag on one allele of the TBP-encoding gene in diploid cells, which is expected to reduce the on-rate of TBP to the TATA (Fig. 4a). Partial depletion was chosen, because full TBP depletion nearly completely abrogated *GAL10* transcription (Extended Data Fig. 5g). Imaging indicated considerable TBP depletion (Extended Data Fig. 2b), but we note that even full TBP depletion was likely incomplete, as evidenced by growth on rapamycin-containing plates of a TBP anchor-away haploid strain (Extended Data Fig. 2a). Partial TBP depletion had a modest effect on *GAL10* transcription (Fig. 4b–d and Extended Data Fig. 5a,b,g–j), likely because the strong *GAL10* TATA ensured sufficient TBP binding even

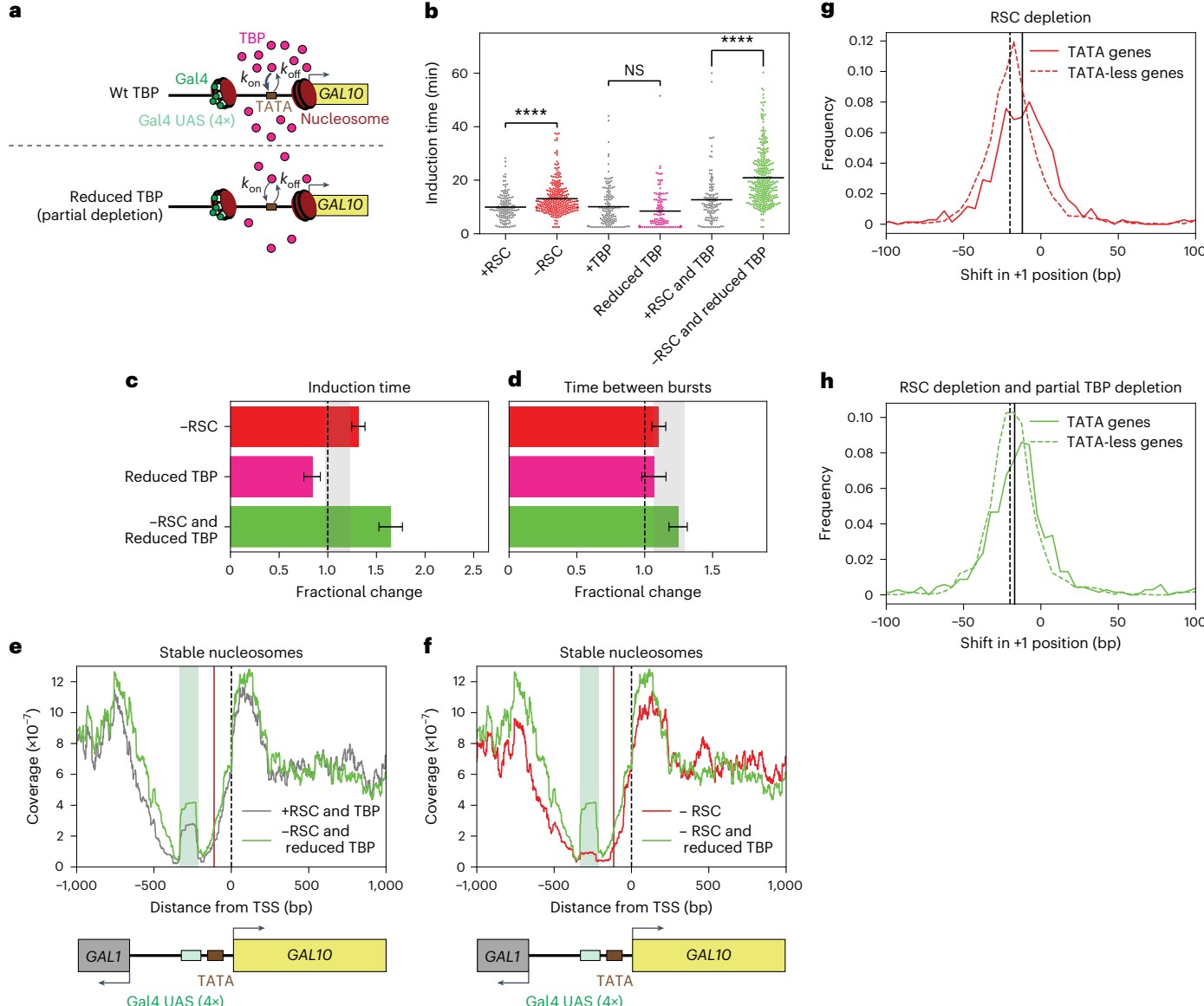

**Fig. 4 | The nucleosomes at the TATA and TSS are redundantly displaced by RSC and TBP to regulate the first burst of *GAL10* transcription. a**, Schematic showing a reduced TBP on-rate after the addition of rapamycin in a diploid yeast strain where one of the two copies of TBP was depleted by anchor-away. **b**, *GAL10* induction time increased upon RSC depletion, did not change upon partial TBP depletion and increased more in simultaneous RSC and partial TBP depletion. Significance determined by two-sided bootstrap hypothesis testing[53]; NS, not significant; ****, $P < 0.00005$. $P$ values: ±RSC, $P < 10^{-15}$; +TBP versus partial TBP 0.092; +RSC and TBP versus −RSC and partial TBP, $P < 10^{-15}$. **c**, The increase in *GAL10* induction time when simultaneously depleting RSC and partial TBP was larger than expected based on their individual depletions. Gray bar, expected effect based on dynamic epistasis analysis. **d**, The time between the consecutive bursts of *GAL10* transcription increased as expected in the double depletion of RSC and partial TBP. Gray bar, expected effect based on dynamic epistasis

analysis. Data in **c** and **d** are presented as the fractional change based on mean values ± s.d. based on 1,000 bootstrap repeats. **e**, MNase-seq analysis of stable nucleosomes in the *GAL10* promoter region showed increased coverage around the TSS and TATA when RSC and partial TBP were depleted. **f**, MNase-seq analysis of stable nucleosomes in the *GAL10* promoter region showed higher coverage around the TATA and TSS when depleting RSC and partial TBP than when depleting only RSC. **g**, Histogram of shift in +1 nucleosome upon depletion of RSC for both TATA genes and TATA-mismatch genes, showing that TATA-mismatch genes showed a larger shift in the +1 nucleosome than TATA genes. **h**, Histogram of shift in +1 nucleosome upon simultaneous depletion of RSC and partial TBP for both TATA genes and TATA-mismatch genes. Solid and dotted vertical lines in **g** and **h** indicate the median shift in the +1 nucleosome position of the TATA and TATA-less genes, respectively. Plots in **e**–**h** show one representative replicate out of two experiments.

at a reduced TBP concentration. Surprisingly, partial TBP depletion resulted in faster induction than no depletion (Fig. 4b,c and Extended Data Fig. 5a,b), possibly due to interference of the anchor-away tag with TBP function (Extended Data Fig. 6, Methods).

To study how RSC remodeling at the TATA affects TBP–TATA interaction, TBP and RSC were depleted simultaneously. This combined depletion resulted in a synergistic delay of *GAL10* induction

(Fig. 4b,c and Extended Data Fig. 5c,d) and the expected effect on the time between bursts (Fig. 4d). A similar synergistic delay in induction was obtained for the combined SWI/SNF and TBP depletion (Extended Data Fig. 5e,f,k–p). In contrast to the fragile Gal4 UASs nucleosome, which requires remodeling before each burst, remodeling of the nucleosomes around the TATA is required to initiate the first, but not subsequent, bursts of *GAL10* transcription.

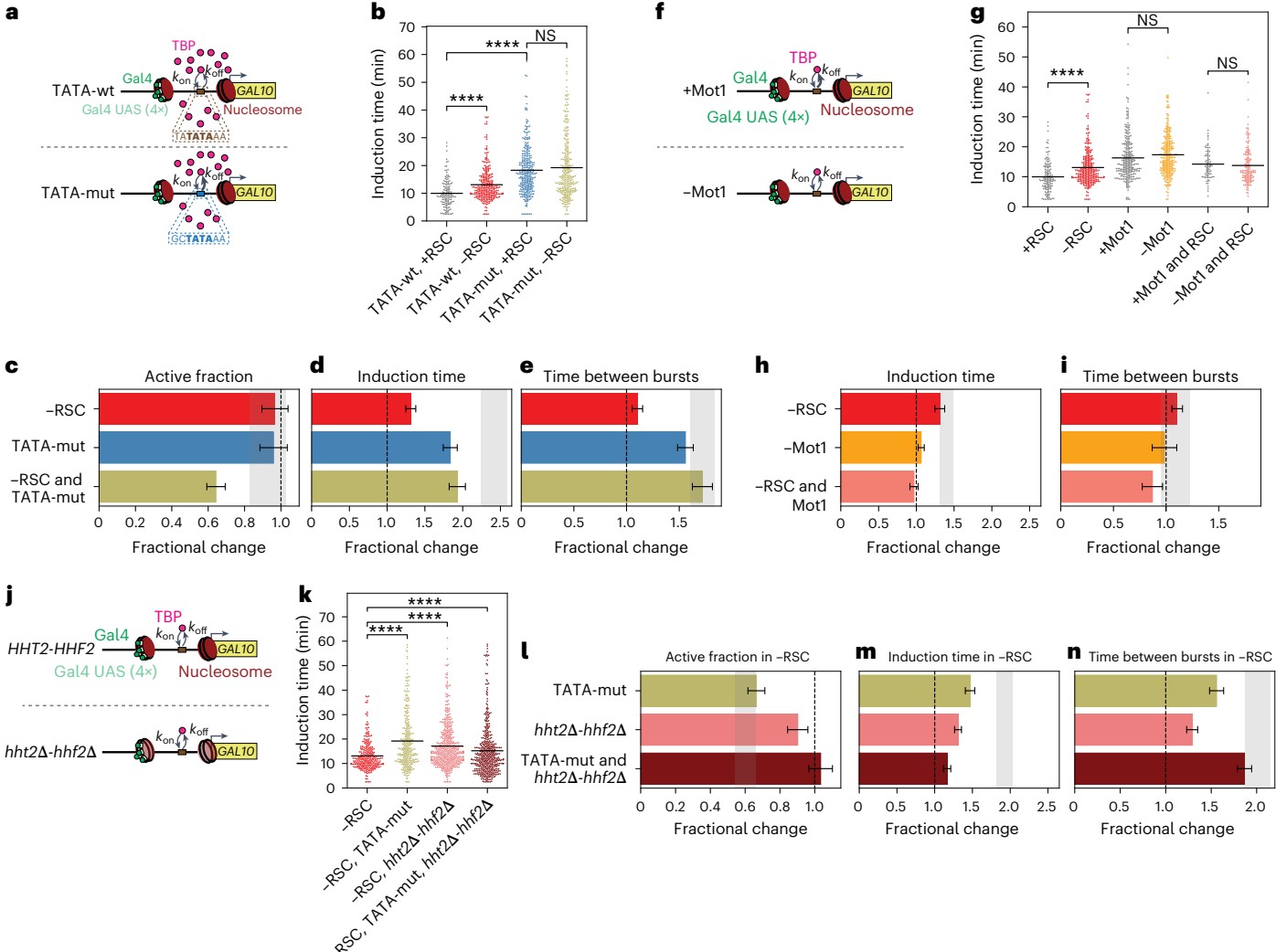

**Fig. 5 | Competition between TATA nucleosome and TBP depends on TBP residence time. a**, Schematic showing expected reduced TBP residence time upon mutating TATA (TATA-mut). **b**, *GAL10* induction time increased upon RSC depletion in TATA-wt, but not in TATA-mut cells. *P* values: TATA-wt, ±RSC, $P < 10^{-15}$; +RSC, TATA-wt versus TATA-mut, $P < 10^{-15}$; TATA-mut, ±RSC, $P = 0.264$. **c**, Upon RSC depletion in TATA-mut, a fraction of cells did not activate *GAL10* transcription. **d**, TATA-mut increased *GAL10* induction time, with no additional effect upon RSC depletion, resulting in a smaller-than-expected effect based on individual perturbations. **e**, The effect on the time between consecutive bursts of *GAL10* transcription was as expected based on individual perturbations. **f**, Schematic showing increased TBP residence time at the TATA upon Mot1 depletion. **g**, *GAL10* induction time increased upon depletion of RSC but not upon Mot1 or RSC and Mot1 depletion. *P* values: ±RSC, $P < 10^{-15}$; ±Mot1, $P = 0.096$; ±RSC and Mot1, $P = 0.6$. **h**, Smaller-than-expected effect of RSC and Mot1 depletion; Mot1 rescued the effect of RSC depletion. **i**, The effect on the time between consecutive bursts of *GAL10* transcription was as expected based on individual perturbations. **j**, Schematic showing reduced histone levels upon *hht2Δ-hhf2Δ*. **k**, In −RSC, *GAL10* induction time is increased upon TATA-mut or upon *hht2Δ-hhf2Δ*, which is partially rescued by their combination. *P* values: −RSC versus −RSC and TATA-mut, $P < 10^{-15}$; −RSC versus −RSC and *hht2Δ-hhf2Δ*, $P < 10^{-15}$; −RSC versus −RSC and TATA-mut and *hht2Δ-hhf2Δ*, $P < 10^{-15}$. **l**, The inactive population in TATA-mut in −RSC is rescued by additional *hht2Δ-hhf2Δ*. **m**, TATA-mut in −RSC increased induction time, but this was rescued by additional *hht2Δ-hhf2Δ*. **n**, The effect on the time between consecutive bursts of *GAL10* transcription was as expected based on individual perturbations. **c**−**e**, **h**−**i** and **l**−**n**, Change in active fraction is the fractional change based on number of active and inactive cells ± propagated statistical errors in these numbers. Change in other parameters is fractional change based on mean values ± s.d. based on 1,000 bootstrap repeats. Gray bar, expected effect based on dynamic epistasis analysis. **l**−**n**, Fractional changes are calculated relative to −RSC cells. **b**, **g** and **k**, Significance determined by bootstrap two-sided hypothesis testing[53]; NS, not significant; ****, $P < 0.00005$.

## Nucleosome competition by TBP binding

The observed synergy between reduced remodeling at the TATA and lower TBP concentration suggested a role for TBP in affecting nucleosome positions. To test this, MNase-seq was performed in cells depleted of RSC and partially depleted of TBP. Indeed, a larger shift in the *GAL10* TSS nucleosome and more nucleosome density over the TATA were observed in these double-depleted cells than in RSC-only depleted cells (Fig. 4e,f and Extended Data Fig. 7a–f). Thus, in cells with impaired chromatin remodeling, TBP binding at the TATA contributes to positioning of the TSS and TATA nucleosomes.

The function for TBP in nucleosome positioning was supported by genome-wide MNase-seq analysis. For genes with a canonical TATA motif, we observed a smaller shift in +1 nucleosome upon RSC depletion and less coverage at the TATA than for genes lacking the canonical TATA motif (Fig. 4g and Extended Data Fig. 7g). Upon additional partial depletion of TBP, the ability of TBP to compete with nucleosomes is specifically reduced at genes with a strong TATA, but not at TATA-mismatch genes (Fig. 4h and Extended Data Fig. 7h,i). These findings suggest that a strong TBP–TATA interaction is required for TBP to compete with nucleosomes.

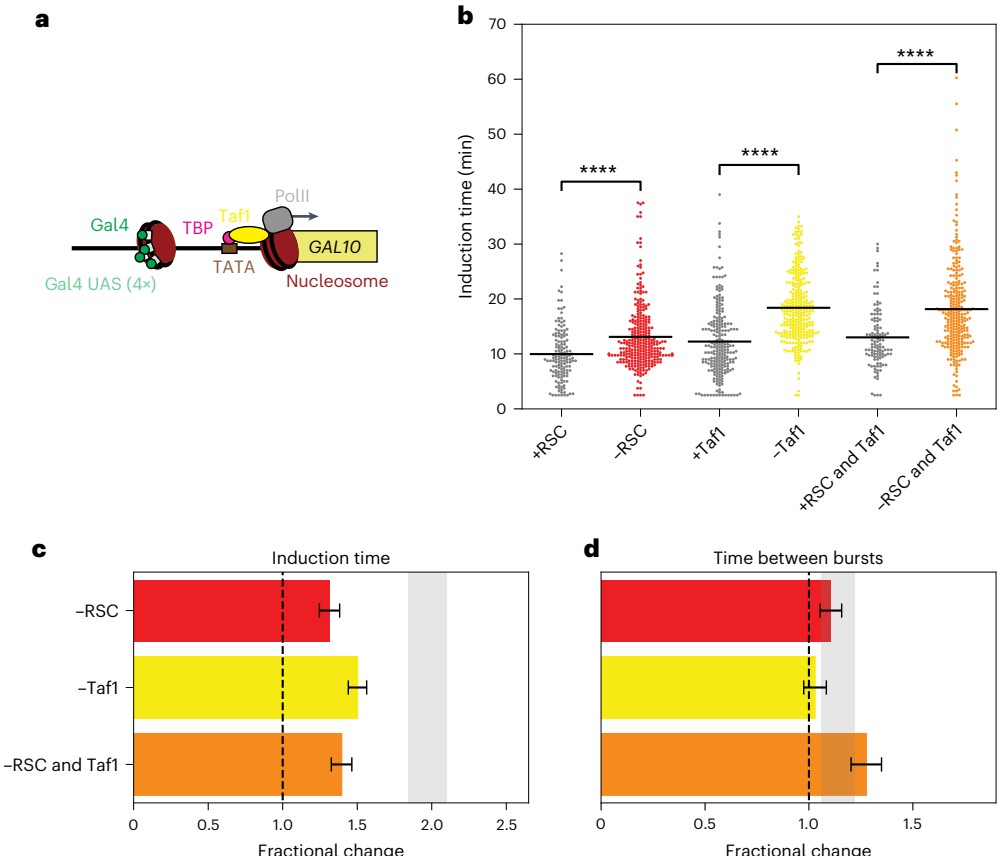

**Fig. 6 | Antagonistic effect on induction time by RSC and Taf1 shows that nucleosomes cannot be competed away by Taf1. a**, Schematic showing Taf1 binding at the +1 nucleosome in the *GAL10* promoter region. **b**, Taf1 depletion and simultaneous RSC and Taf1 depletion increased the induction time of *GAL10*. Significance determined by two-sided bootstrap hypothesis testing[53]; ****, $P < 0.00005$. P values: ±RSC, $P < 10^{-15}$; ±Taf1, $P < 10^{-15}$; ±RSC and Taf1, $P < 10^{-15}$. **c**, Taf1 and RSC depletion had a smaller effect on induction time than expected

based on the individual depletions, indicating that RSC and Taf1 have opposing functions. Gray bar, expected effect based on dynamic epistasis analysis. **d**, Taf1 and RSC depletion had the expected effect on time between consecutive bursts. Gray bar, expected effect based on dynamic epistasis analysis. Data in **c** and **d** are presented as the fractional change based on mean values ± s.d. based on 1,000 bootstrap repeats.

## TBP nucleosome competition depends on the TBP dwell time

To evaluate whether the strength of the TBP–TATA interaction determines the ability of TBP to compete with nucleosomes, we mutated the *GAL10* TATA[38] (Fig. 5a). This mutation reduced the burst duration and intensity (Extended Data Fig. 8a,b,k,l) and corroborated predictions of reduced burst size[38]. In addition, the TATA mutation increased the induction time and time between bursts (Fig. 5b,d,e). Remarkably, a RSC depletion in these TATA-mutant cells resulted in two distinct cell populations, as evidenced by a reduced active fraction (Fig. 5c). One population was not able to activate *GAL10* transcription, likely because the TATA nucleosome could not be remodeled by TBP in the absence of RSC. For the active population, there was no additional effect of RSC depletion in terms of induction time (Fig. 5d) and the time between bursts showed the expected effect (Fig. 5e). These cells likely had no nucleosome covering the TATA and thus did not require nucleosome remodeling. TBP could then bind and initiate transcription, resulting in RSC-independent *GAL10* induction. These results are thus in line with our prediction that a strong TATA is required for TBP to compete with nucleosomes.

The inability of TBP to compete with nucleosomes upon a TATA mutation suggested that TBP nucleosome competition and subsequent transcription activation requires a longer residence time on DNA. To test whether increasing the TBP residence time increases TBP nucleosome competition, we stabilized the TBP–TATA interaction through the depletion of Mot1, the protein facilitating TBP removal[39–41] (Fig. 5f).

Mot1 depletion had a modest transcriptional phenotype (Fig. 5g–i and Extended Data Figs. 2k and 8c–f,m–o), perhaps because the depletion was not complete, as indicated by growth on rapamycin-containing plates and imaging (Extended Data Fig. 2a,b). As predicted, simultaneous depletion of RSC and Mot1 led to a rescue of the effect of RSC depletion on *GAL10* induction time and the time between bursts. Stabilized TBP–TATA interaction thus enhances the nucleosome competition ability of TBP, and is sufficient to activate transcription efficiently, even in the absence of RSC remodeling.

## Reduced histone levels rescue transcription dynamics

If TBP competes with nucleosomes, it is expected that impaired TBP binding due to a TATA mutation or reduced TBP levels is rescued if histones are less abundant. To verify this, we analyzed how deleting one of the two H3–H4 gene pairs, *hht2Δ-hhf2Δ* (Fig. 5j) affects bursting in cells with wildtype (wt) or mutant TATA, in the presence and in the absence of RSC. Western blot analysis using a H3K79me3 antibody suggests reduced histone levels (Extended Data Fig. 8t; H3K79me3 is present on 90% of the nucleosomes[42]), even though a Western blot with the H3 antibody did not detect a change in the histone levels. Measurements of transcriptional bursting revealed that *hht2Δ-hhf2Δ* affected several transcriptional bursting parameters (Fig. 5j–n and Extended Data Fig. 8u–v). As hypothesized, the inactive population that was present in the TATA mutant upon RSC depletion (Fig. 5c,l) was absent when histones were also deleted (Fig. 5l). The ability of TBP to compete

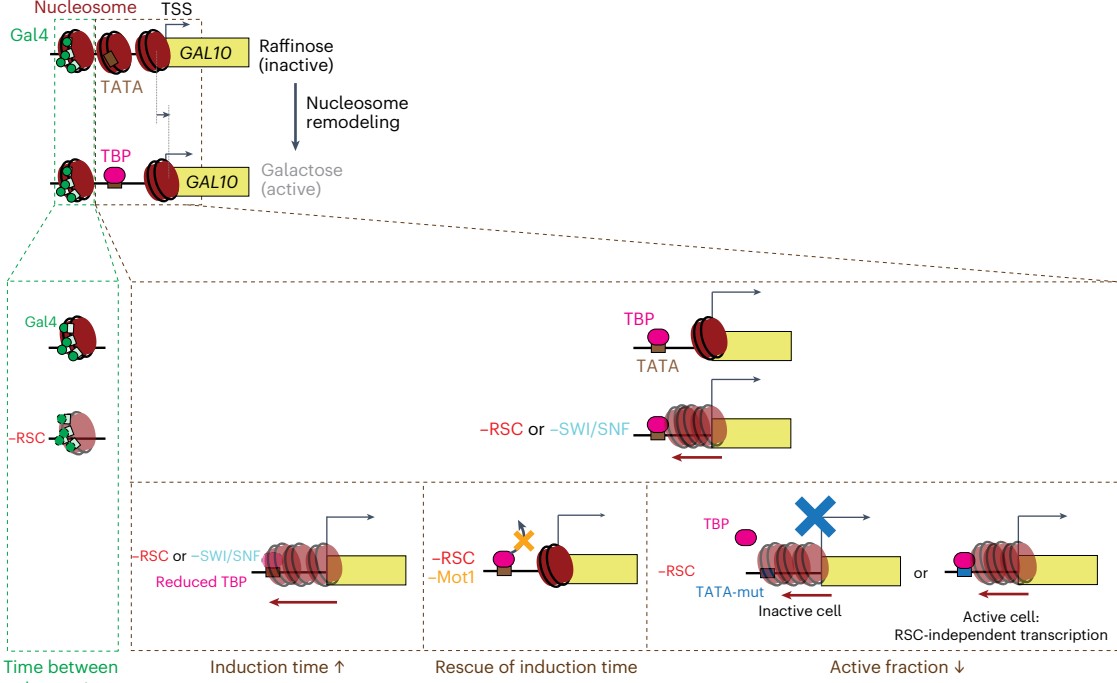

**Fig. 7 | Model showing how the remodeling of different nucleosomes regulates transcriptional bursting at the *GAL10* gene.** Upon induction of the *GAL10* gene, the three promoter nucleosomes are remodeled to allow for transcription activation. Remodeling of the fragile nucleosome at the Gal4 UASs by RSC and Gal4 binding synergistically regulates the time between bursts. The nucleosomes in the region spanning the TATA and the TSS are redundantly displaced by RSC, SWI/SNF and TBP to regulate the first burst of transcription. Nucleosomes that shift into the promoter upon the loss of RSC and/or SWI/SNF are competed away by TBP. In the absence of remodeling by RSC or

SWI/SNF, the partial depletion of TBP reduces the ability of TBP to compete with nucleosomes, such that induction is synergistically slowed down with RSC depletion. The stabilization of TBP by Mot1 depletion results in increased nucleosome competition and rescue of induction time. The mutation of the canonical TATA abolishes the ability of TBP to compete with nucleosomes, resulting in two populations, where either the nucleosome cover the TATA resulting in no activation, or the nucleosome is not covering the TATA resulting in RSC-independent transcription activation.

---

with nucleosomes is thus restored if nucleosomes are less abundant. In addition, both in the absence and in the presence of RSC, the effect of the TATA mutation on several other bursting parameters was partially or fully rescued by *hht2Δ-hhf2Δ* (Fig. 5j,k,m and Extended Data Fig. 8u–w). To verify these results further, we combined partial TBP and RSC depletion with *hht2Δ-hhf2Δ* and found a similar rescue (Extended Data Fig. 8x). The only bursting parameter that did not show a rescue by *hht2Δ-hhf2Δ* was the time between bursts (Fig. 5n and Extended Data Fig. 8w,x), in line with our model that the TBP nucleosome competition regulates the first but not subsequent bursts. Because *hht2Δ-hhf2Δ* is not a conditional perturbation, it is important to note that indirect effects may contribute to the observed effects. Nevertheless, these results agree with the model that TBP binding to the TATA competes with nucleosomes to regulate transcription.

**Taf1 and RSC remodeling act antagonistically at the TSS**
To test if the redundant nucleosome displacement at the TATA by RSC and TBP binding is specific for the TBP–TATA interaction or a general effect from potentially impaired PIC assembly by perturbed TBP–TATA interactions, we measured the effect of Taf1 depletion in combination with RSC depletion. Taf1 is part of the PIC but has no direct interaction with the TATA or the TATA nucleosome, but interacts with the nucleosome around the TSS[43] (Fig. 6a). For Taf1-depleted cells, a delayed *GAL10* induction was observed (Fig. 6b–d and Extended Data Fig. 8g–j,p–r). Rather than the synergistic effect observed between RSC and TBP depletion, we observed a smaller effect on induction time than expected when depleting RSC and Taf1, suggesting opposing roles of RSC and Taf1 in controlling induction time (Fig. 6c). Therefore, in contrast to TBP, Taf1 cannot compete with nucleosomes. Conversely, nucleosome

remodeling of RSC at the TSS acts antagonistically with Taf1 binding to control induction time.

## Discussion
In this study, we use a dynamic epistasis analysis of single and combined perturbations of nucleosome remodeling, the transcription machinery and histones in combination with nucleosome mapping experiments and single-molecule live-cell imaging at the *GAL10* gene in *Saccharomyces cerevisiae*, to uncover how the remodeling of promoter nucleosomes regulates transcriptional bursting. Based on our findings, we propose a model (Fig. 7) where different promoter nucleosomes have specialized roles in controlling transcription dynamics. Specifically, a fragile nucleosome covering the Gal4 UASs is repeatedly remodeled by RSC, in a manner redundant with Gal4 binding, which regulates the start of each transcriptional burst. Additionally, the nucleosomes around the TATA and TSS are positioned by an interplay of RSC, SWI/SNF and TBP to control the first burst of *GAL10* transcription.

**Nucleosome at the Gal4 UASs**
Similar to observations of the TF Ace1 at the CUP1 array[8], dynamic remodeling of the fragile nucleosome at the Gal4 UASs by RSC and Gal4 binding is required to allow each burst of transcription to start. The synergy between reduced Gal4 concentration and RSC depletion indicates that Gal4 binding can substitute for RSC in configuring the nucleosome to a state allowing transcription, in a manner similar to what has been observed for general regulatory factors, such as Reb1 or Abf1, although at most genes these general regulatory factors act independently of RSC rather than redundantly, as we observe for Gal4[19,23]. Mechanistically, Gal4 may trap the nucleosome in a partially unwrapped

state, as was shown in vitro[30]. The binding of a gene-specific TF and RSC remodeling thus cooperate to enable efficient TF binding at each burst of transcription.

## Nucleosome at the TATA

Our study uncovers a role for TBP in competing with nucleosomes covering the TATA. Although TBP depletion in the presence of RSC has a minor effect on nucleosome positioning only at highly expressed genes[19,44], we show that the effect of TBP on nucleosome displacement becomes more prominent in conditions where nucleosomes cover the TATA, such as after the depletion of RSC or SWI/SNF. Specifically, the simultaneous depletion of TBP and RSC results in a larger increase in the nucleosome density around the *GAL10* TATA than single RSC depletion (Fig. 4f). This role for TBP in nucleosome positioning appears more important for genes with a canonical TATA, possibly by supporting longer TBP residence times (Fig. 4 and Extended Data Fig. 7). The ability of TBP to compete with nucleosomes allows for remodeler-independent *GAL10* promoter activation, explaining why a substantial level of *GAL10* transcription is still observed in the absence of RSC and SWI/SNF.

It was recently shown that TBP is able to bind stably to nucleosomal DNA[32] but cannot efficiently recruit the PIC and activate transcription when nucleosome bound[32,45]. Our experiments where the TBP–TATA interaction is destabilized by a TATA mutation or stabilized by Mot1 depletion reveal that the ability of TBP to compete with nucleosomes may depend on the residence time of TBP at TATA. Although Mot1 depletion could, in principle, rescue *GAL10* transcription through its regulation of antisense transcription[39,46,47] rather than by increasing the TBP dwell time[39,48,49], we find this unlikely, because *GAL10* antisense transcription does not affect *GAL10* bursting[50]. Rather, we envision a passive competition mechanism where the binding of TBP may perturb the stability of the nucleosome by recruiting PIC components to partially unwrapped nucleosomes intermediates that arise from spontaneous nucleosome breathing[51]. If the residence time of TBP is long enough, the successive binding of multiple PIC components may eventually lead to nucleosome eviction, resulting in a nucleosome-free TATA needed for complete PIC assembly. Alternatively, the bending of the DNA that is introduced by the binding of TBP may change the nucleosome positioning energy landscape[52]. Longer TBP binding may increase the probability of downstream nucleosome movement to energetically more favorable sites. Moreover, because TBP–TATA binding is one of the most stable interactions within the transcription assembly, with a residence time of several minutes[31,46], we envision that, once bound, TBP can maintain a nucleosome-free TATA, facilitating the initiation of multiple consecutive bursts. This is in line with our observed synergy between TBP and RSC in controlling the first, but not consecutive, transcriptional bursts.

## Nucleosome at the TSS

The observed nucleosome competition is specific for TBP, rather than being a common mode of action of all PIC components, because Taf1 depletion does not show the same synergy with RSC depletion as TBP depletion. Even though Taf1 interacts stably with the TSS nucleosome in yeast extracts[43], Taf1 and other PIC components interact with chromatin only for a few seconds in vivo[31], contradicting the stable chromatin engagement that may be needed to passively compete away nucleosomes such as we propose for TBP[28]. In addition, we observe a lower induction time than is expected for the simultaneous depletion of Taf1 and RSC, indicating that Taf1 and RSC have opposing functions. In support, recent single-molecule tracking measurements of PIC components revealed that Taf1 binding to chromatin becomes more stable upon RSC depletion, suggesting that RSC promotes TFIID turnover[31]. Our data thus indicate that RSC, SWI/SNF and TBP redundantly are able to displace nucleosomes around the TATA and that RSC inhibits stable Taf1 binding around the TSS nucleosome.

Overall, dynamic epistasis analysis provides a detailed mechanistic insight into how nucleosome remodeling acts in combination with TFs and PIC assembly to control the kinetics of transcriptional bursting. In particular, at the yeast *GAL10* gene and other TATA-containing genes, a role for TBP in competing with nucleosomes in vivo is uncovered, which together with chromatin remodelers enables efficient PIC assembly and transcription initiation. Moreover, our work forms a framework for future studies to understand how transcriptional bursting is regulated by the interplay of different transcriptional regulators, of different complex submodules and of complexes that vary in subunit composition or protein isoforms.

## Online content

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

## Methods

### Yeast strains and plasmids

All strains were derived from BY4741 and BY4742 anchor-away background strains[54]. FRB-tags for anchor-away were introduced either by transformation with a polymerase chain reaction (PCR) product containing the FRB-yEGFP1-hphMX4 cassette (pTL100) or the FRB-mScarlet-hphMX4 cassette (pTL329). Alternatively, a CRISPR–Cas9-based approach was used[55], and strains were transformed using a plasmid expressing Cas9, a guide RNA and double-stranded PCR repair template from the same plasmids, followed by removal of the Cas9 plasmid by 5-fluoroorotic acid selection. PP7 loops were introduced by transformation with a PCR product containing the PP7 loop cassette and loxP-kanMX-loxP (pTL031), and subsequent removal of the kanMX marker by CRE recombinase expression (pTL014 or pTL191). PP7-coat protein was integrated at *URA3* by transformation of a PacI-digested single-integration plasmid (pTL174)[56]. TATA-mut-int238 was introduced in the *GAL10* TATA by the CRISPR–Cas9 approach described above using a single-stranded oligo as the repair template. The *GAL4/gal4Δ* strain was created by mating a BY4741 wt *GAL4* anchor-away strain with a BY4742 anchor-away *gal4Δ* haploid strain that was constructed by the CRISPR–Cas9 approach described above, using a single-stranded oligo as repair template. The *GAL4-3V5/GAL4-3V5* strain was created by mating a BY4741 and a BY4742 anchor-away *GAL4-3V5* strain that were constructed by the CRISPR–Cas9 approach described above using a repair template created by a PCR product from genomic DNA from a *GAL4-3V5* strain (YTL1446) and verified using Sanger sequencing. The *GAL4-3V5/gal4Δ* strain was created by mating the same BY4741 *GAL4-3V5* strain with a BY4742 *gal4Δ* haploid strain. The *hht2Δ-hhf2Δ* strains were constructed by the CRISPR–Cas9 approach described above using a single-stranded oligo as the repair template. For all strains, at least two replicates were constructed independently, which were verified by PCR, growth plates (Extended Data Fig. 2a), microscopy (Extended Data Fig. 2b) and, if applicable, sequencing and smFISH (Extended Data Fig. 2c). All strains, plasmids and oligos used in this study are listed in Supplementary Tables 1, 2 and 3, respectively. The yeast strains and plasmids are available on request.

### Live-cell imaging of transcription dynamics

Live-cell imaging of transcription dynamics was performed as previously described in detail[4,57] with minor modifications. In brief, cells were treated with 7.5 μM rapamycin or dimethyl sulfoxide (DMSO) for 60 min and subsequently imaged at mid-log (optical density (OD$_{600nm}$) 0.2–0.4) on a coverslip with an agarose pad consisting of 2% agarose and synthetic complete medium containing 2% galactose and 7.5 mM DMSO or rapamycin. Imaging was performed on a setup consisting of an AxioObserver inverted microscope (Zeiss), an alpha Plan-Apochromat ×100 numerical aperture (NA) 1.46 oil objective, an sCMOS ORCA Flash 4v.3 (Hamamatsu) with a 475–570 nm dichroic (Chroma), 570 nm longpass beamsplitter (Chroma) and 515/30 nm emission filter (Semrock), a UNO Top stage incubator (OKOlab) at 30 °C and light emitting diode (LED) excitation at 470/24 nm (Spectra X, Lumencor) at 20% power and an neutral density (ND) 2.0 filter, resulting in a 62 mW cm$^{-2}$ excitation intensity. Widefield images were recorded for 1 h at 15 s intervals, with *z*-stacks (nine slices, Δ*z* 0.5 μm) and 150 ms exposure using Micro-Manager software[58]. For each condition, at least two and often three replicate datasets were acquired with at least 80 cells in total.

### Microscopy of anchor-away nuclear depletion

Before measuring the transcription dynamics, proper nuclear depletion was ensured in each sample by imaging of FRB-mScarlet. Imaging was performed on the setup described above but with 475–570 nm dichroic (Chroma), 570 nm longpass beamsplitter (Chroma) and 600/52 nm emission filter (Semrock) and LED excitation at 550/15 nm (Spectra X, Lumencor) at 100% power and ND 2.0 filter, resulting in 413.0 mW cm$^{-2}$ excitation intensity. A single widefield image was recorded as a *z*-stack (nine slices, Δ*z* 0.5 μm) using either 150 ms exposure (for YTL1178, YTL1179, YTL1281, YTL1309) or 500 ms exposure (for YTL1505, YTL1506, YTL1508, YTL1510, YTL1448, YTL1450, YTL1470, YTL1591, YTL1626, YTL1750, YTL1752, YTL1751).

### Analysis of transcription dynamics

For the analysis of the transcription dynamics imaging data, a similar approach was used to that previously described[4]. All analysis was implemented as custom-written Python software (https://github.com/Lenstralab/livecell). First, the images were corrected for *xy*-drift in the stage using an affine transformation on the maximum intensity projection. Next, the cells were segmented using Otsu thresholding and watershedding. The intensity of the TS was calculated by fitting a two-dimensional Gaussian mask after the local background subtractions as described previously[59]. Initially, a threshold of six times the standard deviation (s.d.) of the background was used. For frames where no TS was detected, a second fit was made in the vicinity of the high-intensity spots detected in that cell, using a threshold of four times the s.d. of the background. For frames where still no TS was detected, the intensity was measured at the location of the previous frame where a TS was detected. The tracking within each cell was inspected visually, and the endpoint of each trace was manually set at the last frame with a visible TS. Cells without a TS, cells that were segmented incorrectly, and cells in which the track contained tracking errors were excluded from analysis.

Binarization was performed using a threshold set at five times the s.d. of the background, determined for each cell by fitting a Lorentzian distribution to intensities measured at four points at a fixed distance from the TS in each frame. This threshold reliably distinguished on and off periods at the single-transcript level. Subsequently, the binarization was improved by removing the bursts that lasted a single frame and merging bursts that were separated by a single frame. From these binarized traces, the burst durations, time between bursts and induction time were directly calculated. The burst intensity was measured as the average intensity of all frames in which the cell was on. The fraction of active cells was determined by the manual scoring of whether the cells show a TS during the 1 h acquisition. Reported values for burst duration, time between bursts, induction time and burst intensity were determined by bootstrapping with 1,000 repetitions. Reported error bars are the s.d. from the same bootstrap. Error bars in the number of active and inactive cells are given by the square root of the number of cells. To determine whether the active fractions are significantly different between conditions, a two-sided Fisher's exact test was used. For the other parameters, we have used bootstrap hypothesis testing using equation (4) from ref. [53] to determine the achieved significance level.

### Dynamic epistasis analysis

For the dynamic epistasis analysis, the fractional change in each parameter of the transcriptional bursting was determined as the ratio between the bootstrap mean of this parameter in the perturbed population and the unperturbed population. For the effect of the *hht2Δ-hhf2Δ* perturbations (Fig. 5 and Extended Data Fig. 7), the fractional change in the presence and absence of RSC was calculated relative to +RSC and −RSC, respectively. The error bars were calculated from the same bootstraps and propagated under the assumption that the measurements are independent between conditions. To calculate the expected effect of a double perturbation on each parameter, fractional changes of the individual perturbations are multiplied, analogous to the way phenotypic growth effects caused by pairwise genetic interactions are assessed[36]. The error bars are calculated by error propagation of the errors of individual perturbations.

### Fitting of induction time distributions

To determine whether gene induction depends on a single or multiple rate-limiting steps, a least-squares fit was performed on the histogram of the distribution of induction times, with a binsize of 1 min.

The following parameterization of the probability density function of the Gamma distribution was used:

$$f(x, k, \theta) = A \frac{1}{\theta^k \Gamma(k)} x^{k-1} e^{-\frac{x}{\theta}}$$

Here, $\Gamma(k)$ is the Gamma function, defined as: $\Gamma(k) = \int_0^\infty t^{k-1} e^{-t} dt$

Here $t$ is the variable of integration. The amplitude parameter $A$ was added because there is a dead-time between addition of galactose and actual start of the image acquisition. Free parameters in the fit are $A$ (with lower bound 1 and initial guess 10), $k$ (with lower bound 0.0001 and initial guess 10 for non-memory induction or 1.0001 for re-induction conditions) and $\theta$ (with lower bound 0 and initial guess 1). The scale parameter $k$ is a measure for the number of rate-limiting steps.

### Testing for subpopulations

Dynamic epistasis analysis relies on the use of the bootstrap mean to describe each parameter in a given condition. This is valid if the cells behave as a single population, but masks the potential effects of the specific subpopulations. To determine whether there are subpopulations with different behavior, we tested whether the shapes of these distributions were well described by a theoretical distribution (Extended Data Fig. 6). For induction time a Gamma function was fit as above. For the time between bursts and burst duration a Gamma function was fit (with $A = 1$, $k$ with lower bound 1 and initial guess 1.0001 and $\theta$ with lower bound 0 and initial guess 1). Theoretically, the burst duration should be described by the sum of a deterministic time (the elongation time for *GAL10*) and an exponential distribution describing the initiation kinetics, but this was approximated with a Gamma distribution for simplicity. For the burst intensity a log-normal distribution was fit:

$$f(x, \mu, \sigma) = \frac{1}{x\sigma\sqrt{2\pi}} e^{-\frac{(\ln(x) - \mu)^2}{2\sigma^2}}$$

Free parameters in the fit are $\sigma$ (with lower bound 0 and initial guess 0.6) and $\mu$ (unbounded with initial guess 5.5). The goodness of all fits was determined by calculating the $R^2$ between the data and the fit according to the following formula, where $y_1...y_n$ are the observed values, with an average value of $\bar{y}$, and $f_1...f_n$ are the fitted values:

$$R^2 = 1 - \frac{\sum_i (y_i - f_i)^2}{\sum_i (y_i - \bar{y})^2}$$

For the time between bursts, Gamma fits consistently showed $k$ between 1 and 1.6 with $R^2 > 0.8$ (Extended Data Fig. 6d), indicating these distributions are approximately described by single populations with an exponential distribution. The fits to the burst duration and burst intensity distributions showed high $R^2$ values (>0.97 and >0.94, respectively, Extended Data Fig. 6c,e). In contrast, for the induction time distributions, we found four experiments with $R^2 < 0.8$ (Extended Data Fig. 6b), which could be caused by the presence of multiple populations. However, careful analysis of these distributions did not reveal signs of subpopulations. We noted that three of the low $R^2$ experiments describe cells where TBP was tagged for depletion. Together with the finding that TBP tagging resulted in unexpected faster induction (Fig. 4) and showed deviating shapes of the induction time distribution, this suggested that the anchor-away tag may partially interfere with TBP function. For the fourth experiment, *GAL4/gal4Δ* upon RSC depletion, the low $R^2$ value appeared to be caused by data sparsity, even though we included 181 cells. This data sparsity also appeared to cause inconsistencies in the $k$ values of the Gamma fits to the induction times (Extended Data Fig. 6b), preventing proper interpretation. In addition, the inconsistent $k$ values could arise because the Gamma distribution assumes different rate-limiting steps with equal rates, which may not be valid in all conditions. Overall, this analysis shows no signs of subpopulations, justifying our dynamic epistasis analysis based on the bootstrap mean.

### smFISH

The smFISH was performed as previously described[4,60] with minor modifications. In brief, yeast cultures were grown to an early mid-log ($OD_{600nm}$ 0.5), treated with either 7.5 µM rapamycin or DMSO for 60 min for anchor-away before fixation with 5% paraformaldehyde (Electron Microscopy Sciences, 15714-S) for 20 min. Then cells are washed three times with buffer B (1.2 M sorbitol and 100 mM potassium phosphate buffer pH 7.5), permeabilized with 300 U of lyticase (Sigma-Aldrich, L2524-25KU) and washed with buffer B. Cells were immobilized on poly-L-lysine-coated coverslips (Neuvitro) and permeabilized with 70% ethanol overnight or for up to 3 days. Coverslips were hybridized for 4 h at 37 °C with hybridization buffer containing 10% dextran sulfate, 10% formamide, 2 × SSC, and 5 pmole probe. Four PP7 probes labeled with Cy3 (for YTL1178, YTL1179, YTL1281 and YTL1309) or Cy5 (for YTL1448, YTL1450, YTL1505, YTL1506, YTL1508, YTL1510, YTL1470, YTL1591, YTL1626, YTL1750, YTL1752, YTL1751) were targeted to the loops, or 48 probes labeled with Quasar670 (for YTL524, YTL525, YTL526, YTL527, YTL528, YTL529) were targeted to coding region of *GAL10* (Supplementary Table 4). Coverslips were washed 2× for 30 min with 10% formamide, 2× SSC at 37 °C, 1× with 2 × salium sodium citrate (SSC) and 1× for 5 min with 1× PBS at room temperature. Coverslips were mounted on microscope slides using ProLong Gold mounting media with DAPI (Thermo Fisher Scientific, P36934).

The imaging was performed on two similar microscope set-ups consisting of an AxioObserver inverted microscope (Zeiss), a Plan-Apochromat ×40 NA 1.4 oil differential interference contrast (DIC) ultraviolet (UV) objective, a ×1.25 optovar, an sCMOS ORCA Flash 4v.3 (Hamamatsu). For Cy3, we used a 562 nm longpass dichroic (Chroma), 595/50 nm emission filter (Chroma) and 550/15 nm LED excitation at full power (Spectra X, Lumencor), with an excitation intensity at the two microscopes of 6.8 W cm⁻² or 8.8 W cm⁻². For Cy5, we used a 660 nm longpass dichroic (Semrock or Chroma), 697/60 nm emission filter (Chroma) and 640/30 nm LED excitation at full power (Spectra X, Lumencor), with an excitation intensity at the two microscopes of 4.9 W cm⁻² or 6.7 W cm⁻². For DAPI, we used either a 410 nm/490 nm/570 nm/660 nm dichroic (Chroma), a 430/35 nm, 512/45 nm, 593/40 nm, 665 nm longpass emission filter (Chroma) or a 425 nm longpass dichroic (Chroma) and a 460/50 nm emission filter (Chroma) and LED excitation at 395/25 nm at 25% power (Spectra X, Lumencor), with an excitation intensity at the two microscopes of either 1.2 W cm⁻² or 1.9 W cm⁻². For each sample, for at least 50 fields-of-view, a z-stack (21 slices, Δz 0.3 µm) was recorded for DAPI and Cy3 or Cy5, using 25 ms exposure for DAPI and 250 ms exposure for Cy3 and Cy5 using Micro-Manager software.

### Analysis of smFISH

Images were analyzed using custom-written Python software (https://github.com/Lenstralab/smFISH). Here, the cells and nuclei were segmented using Otsu thresholding and watershedding. The spots were localized by fitting a three-dimensional Gaussian mask after local background subtraction[59]. Cells in which no spots were detected were excluded from further analysis, because visual inspection indicated these cells were not properly segmented or not properly permeabilized, such that smFISH probes did not enter the cells. For each cell, the TS was defined as the brightest nuclear spot and the number of RNAs at each TS was determined by normalizing the intensity of each TS to the median fluorescent intensity of the cytoplasmic RNAs detected in all cells. Cells with fewer than five RNAs at the TS were classified as inactive, and cells with five or more RNAs at the TS were classified as active cells. Subsequently, the fraction of active cells and the mean number of RNAs at the TSs of active cells were determined. For each condition, at least three replicate experiments were performed with in total at least 5,000 cells, and the average value and standard error of the mean were determined for both the active fraction and the number of RNAs at the TSs of active cells. The fractional changes of these parameters

upon nuclear depletion of indicated factors were determined from these mean values.

For the classification of cells into G1, S and G2 cell-cycle stages, the sum of the nuclear DAPI intensity in each cell is calculated from a maximum intensity projection. Subsequently, a histogram of all nuclear DAPI intensities (with 50 equally spaced bins) is fit with a Gaussian mixture model consisting of two peaks. Cells are classified as G1 stage if they are in a window of (s.d.$_1$, 0.75 × s.d.$_1$) around the center of the first peak, as G2 stage if they are in a window of (0.5 × s.d.$_2$, 1.5 × s.d.$_2$) around the center of the second peak and as S stage if they are in between the two peaks, where s.d.$_1$ and s.d.$_2$ are the s.d. of the first and second peak, respectively. Fractional changes in active fraction and number of RNAs at the TSs of active cells are determined as described above for each cell-cycle stage separately.

## MNase-seq

The preparation and analysis of mono-nucleosomal DNA was performed as described previously[4,54] with minor modifications. Briefly, cells were grown in SC + 2% raffinose or SC + 2% galactose from OD 0.3 to OD 0.75 and then treated with 7.5 μM rapamycin or DMSO for 60 min. Then, cells were fixed in 1% paraformaldehyde, washed with 1 M sorbitol, treated with spheroplasting buffer (1 M sorbitol, 1 mM β-mercaptoethanol, 10 mg ml$^{-1}$ zymolyase 100T (US Biological, Z1004.250)) and washed twice with 1 M sorbitol. Spheroplasted cells were treated with 0.01171875 U (low MNase) or 0.1875 U (high MNase) micrococcal nuclease (Sigma-Aldrich, N5386-200UN) in digestion buffer (1 M sorbitol, 50 mM NaCl, 10 mM Tris pH 7.4, 5 mM MgCl2, 0.075% NP-40, 1 mM β-mercaptoethanol, 0.5 mM spermidine) at 37 °C. After 45 min, reactions were terminated on ice with 25 mM ethylenediaminetetraacetic acid (EDTA) and 0.5% sodium dodecyl sulfate (SDS). The samples were treated with proteinase K for 1 h at 37 °C and decrosslinked overnight at 65 °C. Digested DNA was extracted with phenol/chloroform (PCI 15:14:1), precipitated with NH$_4$-Ac, and treated with 0.1 mg ml$^{-1}$ RNaseA/T1. The extent of digestion was checked on a 3% agarose gel. For all conditions, two independent experiments were performed, with similar outcomes, except for the SWI/SNF depletion strain treated with high MNase concentration, where one replicate of the DMSO condition was underdigested. For this condition, only one replicate was used for analysis.

Sequencing libraries were prepared using the KAPA HTP Library Preparation Kit (07961901001, KAPA Biosystems) using 1 μg of input DNA, 5 μL of 10 μM adapter, double-sided size selection before and after amplification using 10 cycles. Adapters were created by ligation of Universal adapter to individual sequencing adapters (Supplementary Table 5). Libraries were checked on a Bioanalyzer High Sensitivity DNA kit (Agilent). Sequencing was performed on a NextSeq550. Paired-end 2 × 75 bp reads were aligned to the reference genome SacCer3 (January 2015) using bowtie2 (ref. [61]) with the settings '−sensitive−end-to-end −3 15 −5 5 -X 1980 −no-contain−no-discordant -p 40 -x'. The data have been deposited in National Center for Biotechnology Information's Gene Expression Omnibus (GEO)[62] and are accessible through GEO Series accession number GSE190737.

## Analysis of MNase-seq

Analysis of MNase-seq data was carried out using custom-written Python software (https://github.com/Lenstralab/MNase_analysis). First, the aligned reads were filtered for length and only reads between 95 and 225 bp were retained for analysis. Subsequently, the read coverage was determined on a chromosome-by-chromosome basis by counting the number of reads covering each base, and normalized to the total coverage on the chromosome. Next, the coverage along each gene was determined using all verified open reading frames in the Saccharomyces Genome Database[63]. TATA and TATA-mismatch genes were identified as 'TATA-containing' and 'TATA-less' as previously described[64]. The coverage in TATA or TATA-mismatch regions was

determined as the sum of the coverage in the 8 bp region spanning the TATA or TATA-mismatch sequence.

For metagene plots, genes were aligned at the +1 nucleosome in unperturbed conditions. To determine the position of the +1 nucleosome for each gene in these unperturbed conditions, the (prenormalization) coverage in a 4,000 bp window around the TSS of each gene was extracted from all experiments performed in DMSO using the high MNase concentration (combining the data for all yeast strains, that is YTL524, YTL525, YTL1306 and YTL1584). If the gene was on the Crick strand, the coverage was flipped to facilitate the alignment of all genes. For each gene, these coverages were subsequently summed and smoothed using a Gaussian filter with a 40 bp window. The minimum of this smoothed coverage was determined, and a peak-calling function was used to detect nucleosome peaks. The −1 nucleosome was defined as the first peak before the coverage minimum, and the +1 nucleosome as the first peak after the coverage minimum. Genes for which fewer than two peaks were detected or for which the −1 or +1 nucleosome was detected more than 1,000 bp away from the TSS were excluded from the analysis. To generate metagene plots, the normalized coverage of all genes in a window of 2,000 bp centered at the location of the +1 nucleosome in unperturbed conditions of that gene was averaged. To generate heatmaps of the log2-fold-change of the coverage upon depletions, genes were sorted by the NDR width as determined by the distance between the −1 and +1 nucleosomes in unperturbed conditions. Subsequently, for each gene, the log2-fold-change between the coverage in each depletion (rapamycin) condition and the average coverage between two replicate experiments in the nondepleted (DMSO) condition was calculated. These data were represented as a heatmap. To determine the shift in the +1 position, the location of the +1 nucleosome was determined in each depletion dataset independently using the same steps as performed on the summed coverage to detect the position of the +1 nucleosome in unperturbed conditions. The shift in +1 nucleosome was then defined as the difference between the +1 nucleosome in depleted conditions and the +1 nucleosome as determined from all unperturbed high MNase datasets.

## Protein detection by immunoblot and antibodies

Yeast cultures were grown to early mid-log (OD$_{600nm}$ 0.5), washed in MilliQ, pelleted and snap-frozen on dry ice. For protein extraction, cells were resuspended in 300 μl MilliQ, incubated with 300 μl 0.2 M NaOH for 7 min at room temperature, centrifuged and resuspended in 500 μl 2× SDS-PAGE sample buffer (4% SDS, 20% glycerol, 0.1 M dithiothreitol (DTT), 0.125 M Tris-HCl pH 7.5 and EDTA-free protease inhibitors). Samples were incubated at 95 °C for 5 min while shaking and centrifuged at 800g for 10 min at 4 °C. Then 20 μl of lysate with loading buffer was run on a NuPAGE 3–8% gradient TAC gel (V5) or 16% polyacrylamide gel (histone H3 and H3K79me3), and transferred to a 0.45 μm nitrocellulose membrane for 4 h (V5) and 2 h (histone H3 and H3K79me3). For blocking, the membrane was washed with tris-buffered saline-tween (TBS-T), incubated with PBS containing 5% milk for 1 h at room temperature and washed briefly with TBS-T. The membrane was incubated with PBS containing 2% milk and primary antibody (1:2,000 for αV5, 1:5,000 for αPgk1 and 1:1,000 for αH3 and αH3K79me3) overnight at 4 °C, washed 3× for 10 min with TBS-T, incubated with 2% mild and secondary antibody (1:5000) for 1 h at room temperature, washed 3× for 10 min with TBS-T and 1× for 10 min with PBS, and imaged using an LI-COR Odyssey infrared imager (Biosciences). Western blot analysis was performed using antibodies against V5 (Thermo Fisher Scientific R960-25, RRID: AB_2556564), Pgk1 (Invitrogen 22C5D8, RRID: AB_2532235), histone H3 (RRID:AB_2631108, a kind gift of the F.v.L. laboratory)[65] and histone H3K79me3 (RRID: AB_2631107, a kind gift of the F.v.L. laboratory)[65]. Secondary antibodies used were IRDye 800CW Goat anti-Mouse IgG 925-32210 Li-COR (RRID AB_2687825), IRDye 800CW Goat anti-Rabbit IgG 926-32211 Li-COR (RRID:AB_621843) and IRDye 680RD Donkey anti-Mouse IgG 925-68072 Li-COR (RRID AB_2814912).

Article

## Growth assay

The growth assay used to assess growth rapamycin- or DMSO-containing plates was performed as described previously[4] with minor modifications. Serial fivefold dilutions of YTL559, YTL658, YTL047, YTL525, YTL524, YTL1306, YTL1281, YTL1391, YTL1413, YTL014, YTL1397, YTL1506, YTL1584, YTL1510, YTL1613, YTL1615, YTL1394, YTL1588, YTL1744, YTL1749, YTL1508, YTL1617, YTL1745, YTL1751 and YTL1747 strains were spotted on Yeast Extract Peptone (YEP) + 2% glucose + 7.5 µM rapamycin, YEP + 2% glucose + DMSO, YEP + 2% galactose + 20 µg µl⁻¹ ethidium bromide + 7.5 µM rapamycin, YEP + 2% galactose + 20 µg µl⁻¹ ethidium bromide + DMSO, and YEP + 2% raffinose + 2% galactose + 40 mM lithium chloride + 0.003% methionine + 7.5 µM rapamycin and YEP + 2% raffinose + 2% galactose + 40 mM lithium chloride + 0.003% methionine + DMSO. The growth was assessed after 3 days at 30 °C.

## Reporting summary

Further information on research design is available in the Nature Portfolio Reporting Summary linked to this article.

## Data availability

The MNase-seq datasets generated during this study are available in the National Center for Biotechnology Information's GEO[62] through GEO Series accession number GSE190737. For MNase-seq, reads were aligned to the reference genome SacCer3 (January 2015). MNase-seq metagene plots were generated using all verified open reading frames in the Saccharomyces Genome Database[63]. TATA and TATA-mismatch genes were identified as 'TATA-containing' and 'TATA-less' as previously described[64]. The microscopy data generated during this study are available from the corresponding author on reasonable request. Source data are provided with this paper.

## Code availability

The software code for analysis of transcription dynamics microscopy data is available at https://github.com/Lenstralab/livecell. The software code for analysis of smFISH microscopy data is available at https://github.com/Lenstralab/smFISH. The software code for analysis of MNase-seq data is available at https://github.com/Lenstralab/MNase_analysis.

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

## Acknowledgements

We thank the Holstege laboratory (Prinses Maxima Centrum, Utrecht, the Netherlands) for strains. We thank A. Balwierz (the Netherlands Cancer Institute, Oncode Institute), L. Joosen (the Netherlands Cancer Institute, Oncode Institute), J. Haarhuis (the Netherlands Cancer Institute), H. Teunissen (the Netherlands Cancer Institute), L. Willems (the Netherlands Cancer Institute) and R. van der Weide (the Netherlands Cancer Institute) for assistance with MNase-seq experiments, W. Pomp (the Netherlands Cancer Institute) for assistance with the analysis software, and J. Meeussen (the Netherlands Cancer Institute) and T. van Welsem (the Netherlands Cancer Institute) for assistance with western blotting. We thank the Research High Performance Computing Facility and the Genomics Core Facility of the NKI for assistance. We thank the members of the T.L.L. and F.v.L. laboratories for helpful discussions and E. de Wit (the Netherlands Cancer Institute) and members of the T.L.L. laboratory for the critical reading of the manuscript. This work was supported by an institutional grant of the Dutch Cancer Society and of the Dutch Ministry of Health, Welfare and Sport, the Netherlands Organization for Scientific Research (NWO, 016.Veni.192.071 (I.B.), ZonMW-TOP 91218022 (F.v.L.) and gravitation program CancerGenomiCs.nl (T.L.L.)), Oncode Institute (T.L.L.), which is partly financed by the Dutch Cancer Society, and the European Research Council (Starting Grant 755695 BURSTREG (T.L.L.)).

## Author contributions

I.B. and T.L.L. conceptualized this study. I.B. developed the methodology. I.B. and T.L.L. developed the software for data analysis. I.B. performed data analysis. I.B. and E.K.K. performed the experiments. I.B. and T.L.L. wrote the draft of the manuscript with input from the other authors. F.v.L. and T.L.L. supervised the study. I.B., F.v.L. and T.L.L. acquired funding.

## Competing interests

The authors declare no competing interests.

## Additional information

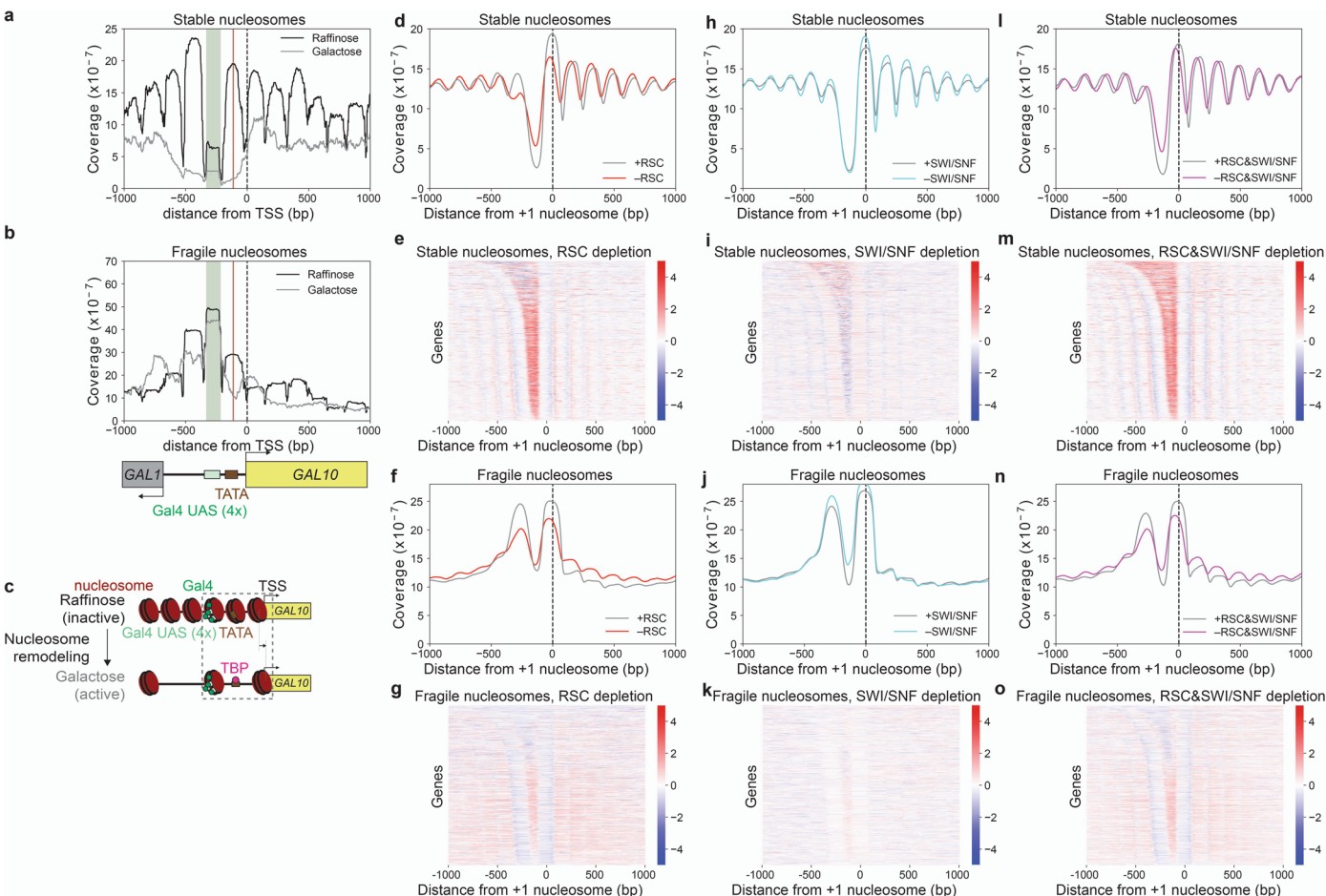

**Extended Data Fig. 1 | Genome-wide changes in nucleosome coverage upon depletion of RSC, SWI/SNF and RSC&SWI/SNF. a**, **b** MNase-seq analysis of (a) stable and (b) fragile nucleosomes in the *GAL10* promoter region in active (galactose) and inactive (raffinose) conditions. **c** Schematic representation of nucleosome remodeling during activation of *GAL10*. Grey box: region with nucleosomes important for regulation of *GAL10* that are discussed in this study. **d**, **f**, **h**, **j**, **l**, **n** Metagene MNase-seq analysis of stable or fragile nucleosomes (as indicated in the figure) upon RSC depletion by anchor-away of Sth1, SWI/SNF depletion by anchor-away of Swi2, or simultaneous depletion of RSC and SWI/SNF, averaged over all genes as annotated by[63], aligned at the location of the +1 nucleosome (black dashed line). **e**, **g**, **i**, **k**, **m**, **o** Heatmap of the log2-fold-change of depleted/nondepleted in nucleosome coverage of stable or fragile nucleosomes (as indicated in the figure) at all genes as annotated by[63], sorted by NDR width and aligned at the location of the +1 nucleosome, upon depletion of RSC, SWI/SNF or RSC and SWI/SNF depletion. Shown is one representative replicate out of two experiments for all except (j) and (k), which is a single replicate experiment.

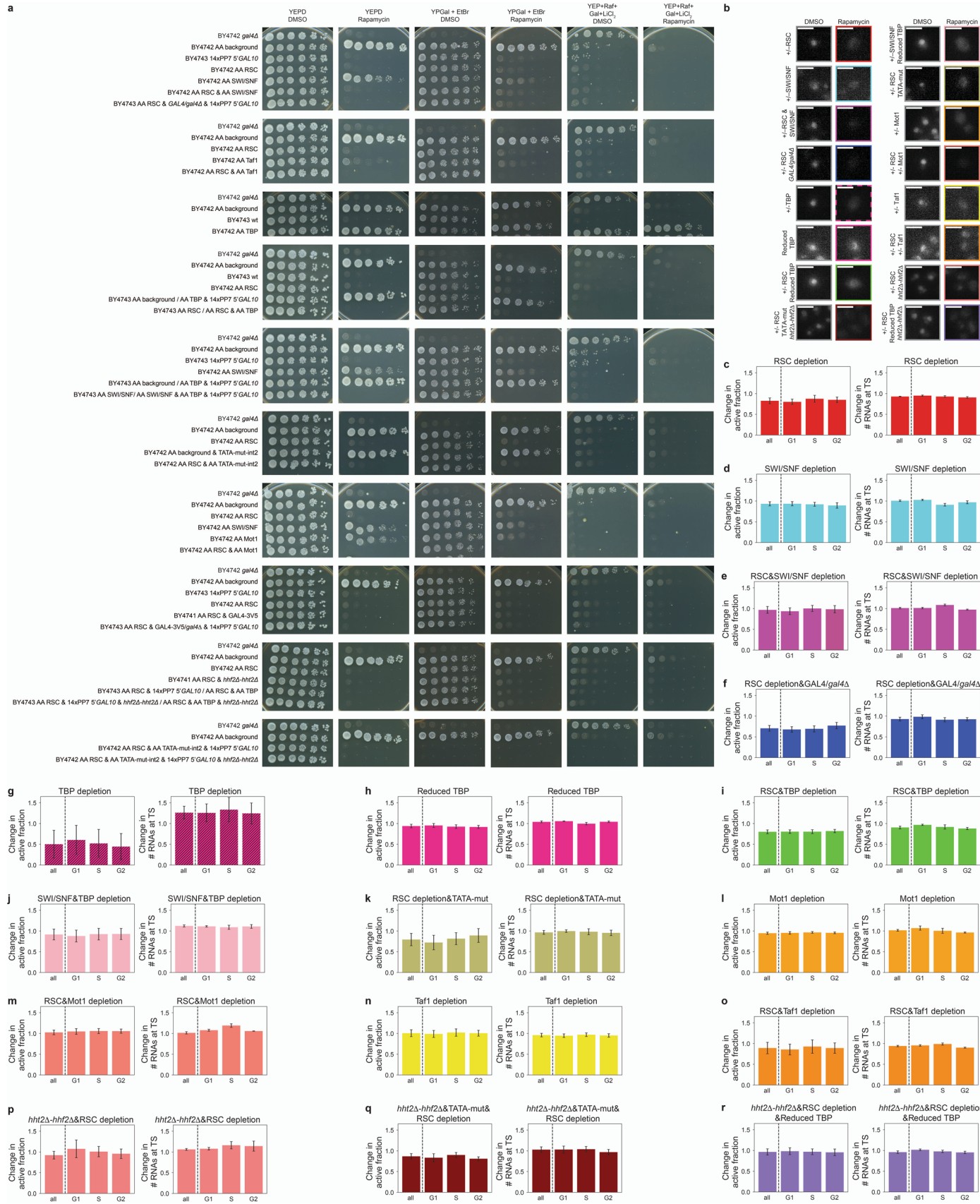

**Extended Data Fig. 2 | See next page for caption.**

**Extended Data Fig. 2 | Verification of strains used for nuclear protein depletions using anchor-away. a** Growth of anchor-away yeast strains used in this study assessed on YEPD, YEP + Galactose + Etidium bromide and YEP + Galactose + Raffinose + Lithium Chloride plates with either DMSO or rapamycin. Shown are (1:5) serial dilutions of cultures, starting at $OD_{600nm}$ of 0.3. **b** Depletion of all indicated factors was verified by imaging cells after 60 min of rapamycin-treatment or control cells treated with DMSO. Shown is 1 typical cell per condition, out of at least 100 cells from three biological replicates. **c-r** No effect of cell-cycle stage on the fraction of (left) active cells and (right) number of RNAs at the TS active cells upon depletion of indicated factors based on smFISH experiments. Active cells defined as cells with 5 or more RNAs at TS. Error bars are SEOM from 3 independent experiments.

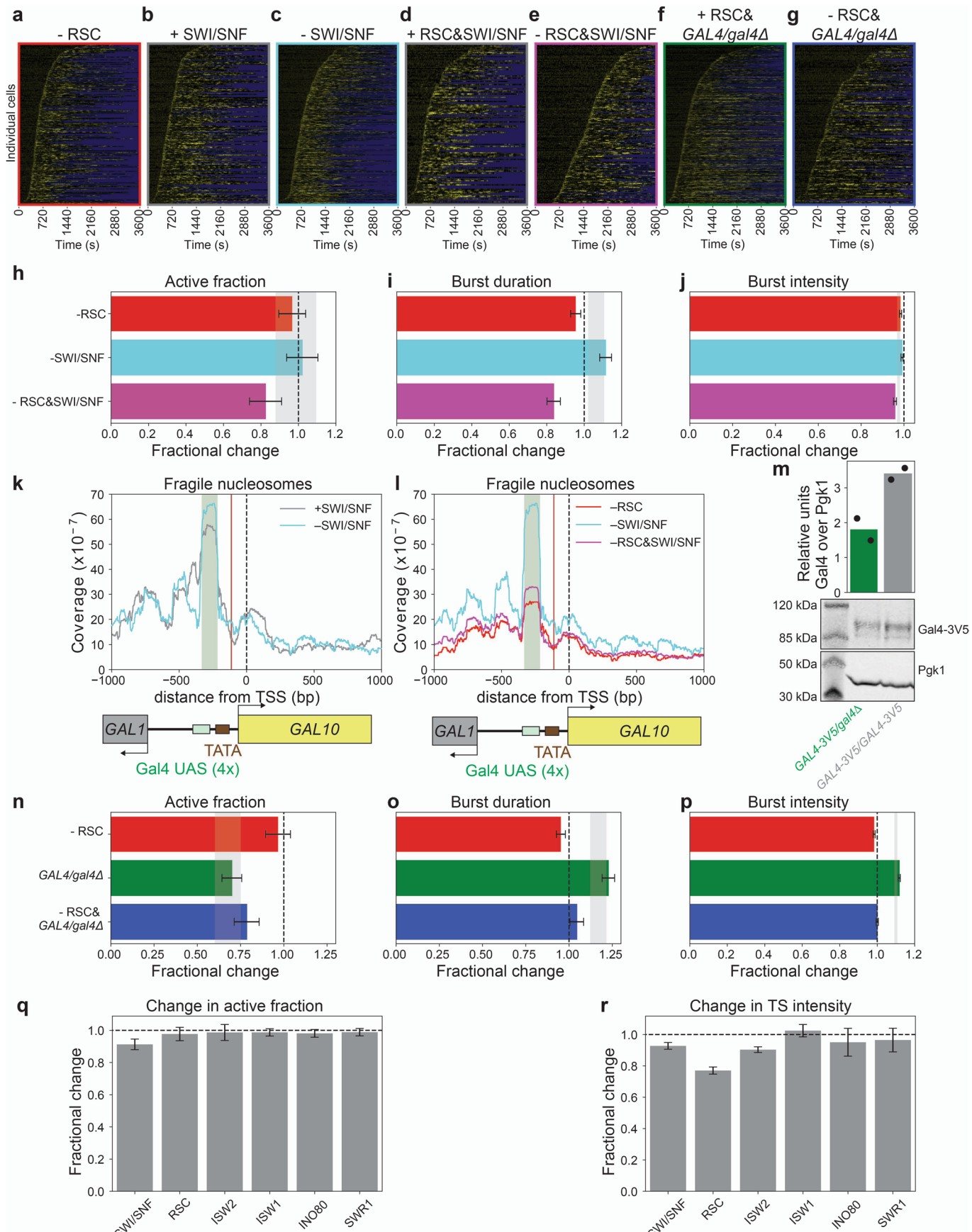

**Extended Data Fig. 3 | See next page for caption.**

**Extended Data Fig. 3 | Change in transcription dynamics and nucleosome coverage upon depletion of chromatin remodeling complexes. a**–**g** Heatmap of TS intensity in (a) N = 292, (b) N = 213, (c) N = 453, (d) N = 151, (e) N = 202, (f) N = 471, and (g) N = 181 cells (rows) in the presence or absence of indicated factors. Yellow: TS fluorescence intensity; blue: region excluded from analysis. **h**–**j** The (h) fraction of cells that activated during 1 hour of imaging, (i) the burst duration and (j) the burst intensity for depletions of RSC and/or SWI/SNF. **k**-**l** MNase-seq analysis of fragile nucleosomes in the *GAL10* promoter region upon (k) SWI/SNF depletion and (l) simultaneous depletion of RSC and SWI/SNF compared to depletion of either RSC or SWI/SNF individually. **m** Western blot analysis with a V5 antibody showed reduced Gal4 expression (0.5 ± 0.1) of *GAL4-3V5/gal4Δ* compared to *GAL4-3V5/GAL4-3V5* cells. Shown is an example blot and quantification over Pgk1 levels of 2 independent experiments. Black dots

indicate individual results of both experiments, bars indicate mean value of the individual experiments. **n**–**p** The (n) fraction of cells that activated during 1 hour of imaging, (o) the burst duration and (p) the burst intensity for depletion of RSC, in *GAL4/gal4Δ* cells and the double perturbation. **q**, **r** Change in (q) fraction of active cells and (r) change in TS intensity of active cells based on smFISH upon depletion of the catalytic subunits of each of the yeast chromatin remodeling complexes. Active cells defined as cells with <5 RNAs at the TS. Error bars are SEOM from 3 independent experiments. (**h**–**j**),(**n**–**p**) Data for active fraction is fractional change based on number of active and inactive cells + / - propagated statistical errors in these numbers. Data for other parameters are fractional change based on mean values + / - standard deviation based on 1000 bootstrap repeats. Grey bar: expected effect based on dynamic epistasis analysis.

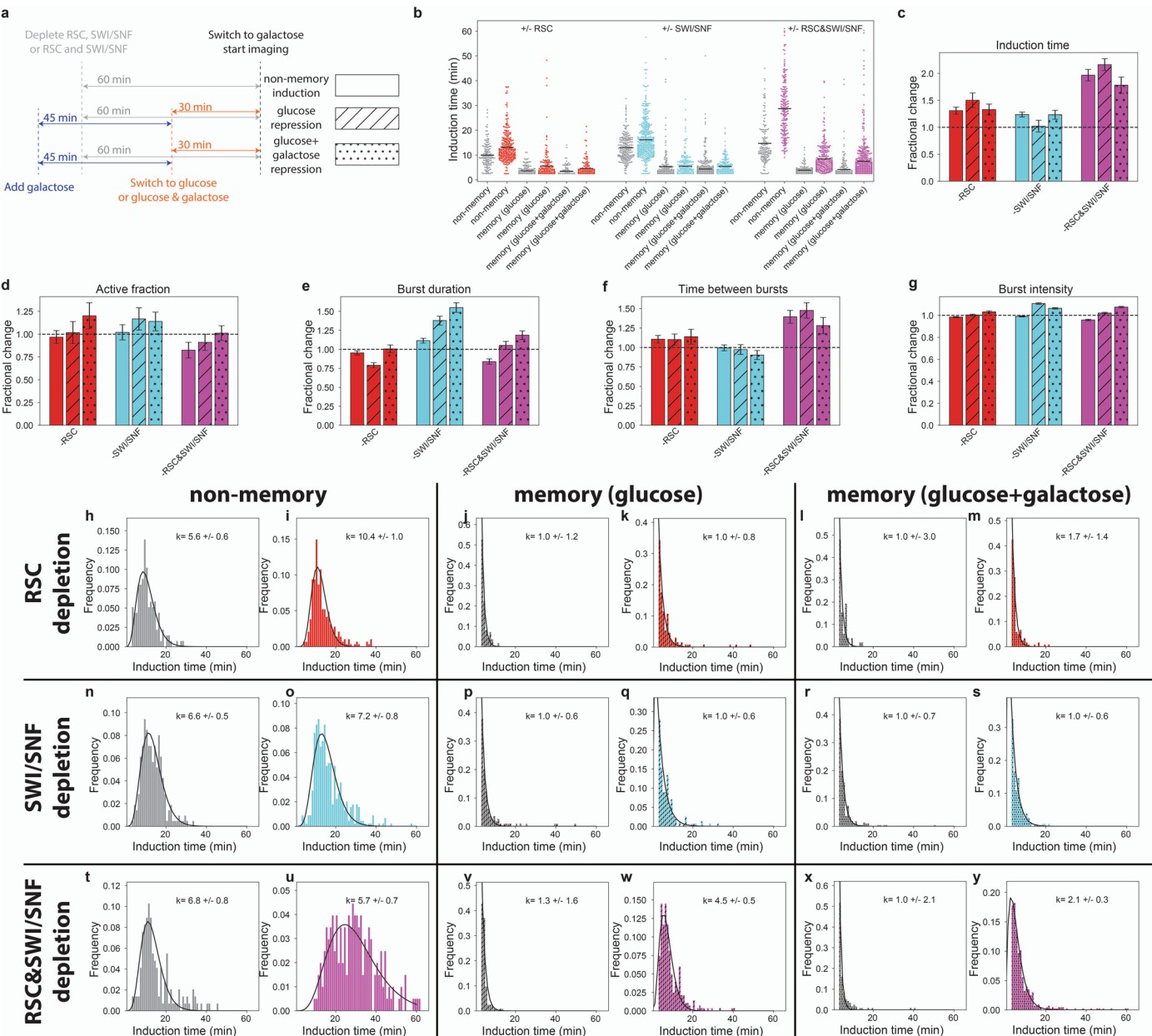

**Extended Data Fig. 4 | Transcriptional re-induction of *GAL10* shows increase in the number of activation steps upon RSC and SWI/SNF depletion.**
**a** Schematic explaining re-induction experiments. For non-memory induction, cells were grown in raffinose. For memory experiments, cells were primed by adding galactose for 45 min and subsequently washed and resuspended in the appropriate repression sugar 30 min before imaging. As in all experiments, rapamycin or DMSO was added for depletion 60 min prior to galactose addition and imaging. **b** *GAL10* induction time in memory and non-memory conditions, upon depletion of RSC, SWI/SNF or both RSC and SWI/SNF. **c–g** Change in (c) induction time, (d) active fraction, (e) burst duration, (f) time between bursts and (g) burst intensity upon depletion of RSC, SWI/SNF or both RSC and SWI/SNF. Data in (c) and (e)-(g) are presented as the fractional change based on mean

values +/- standard deviation based on 1000 bootstrap repeats. Data in (d) is presented as the fractional change based on number of active and inactive cells +/- propagated statistical errors in the number of active and inactive cells. **h–y** Distribution of induction time in memory or non-memory conditions as indicated on top in presence (grey) or absence (red, cyan or magenta) of remodeler indicated on the left. Black line: least-squares fit with Gamma distribution; inset: shape parameter *k* obtained from fit. Error obtained from least-squares fit. A *k*-value not significantly different from 1 indicates a single rate-limiting step, while *k* > 1 indicates multiple rate-limiting steps. Without remodeler depletion, re-induction after memory with glucose shows a single rate-limiting step (v),(x), which increases to multiple rate-limiting steps after combined RSC&SWI/SNF depletion (w),(y).

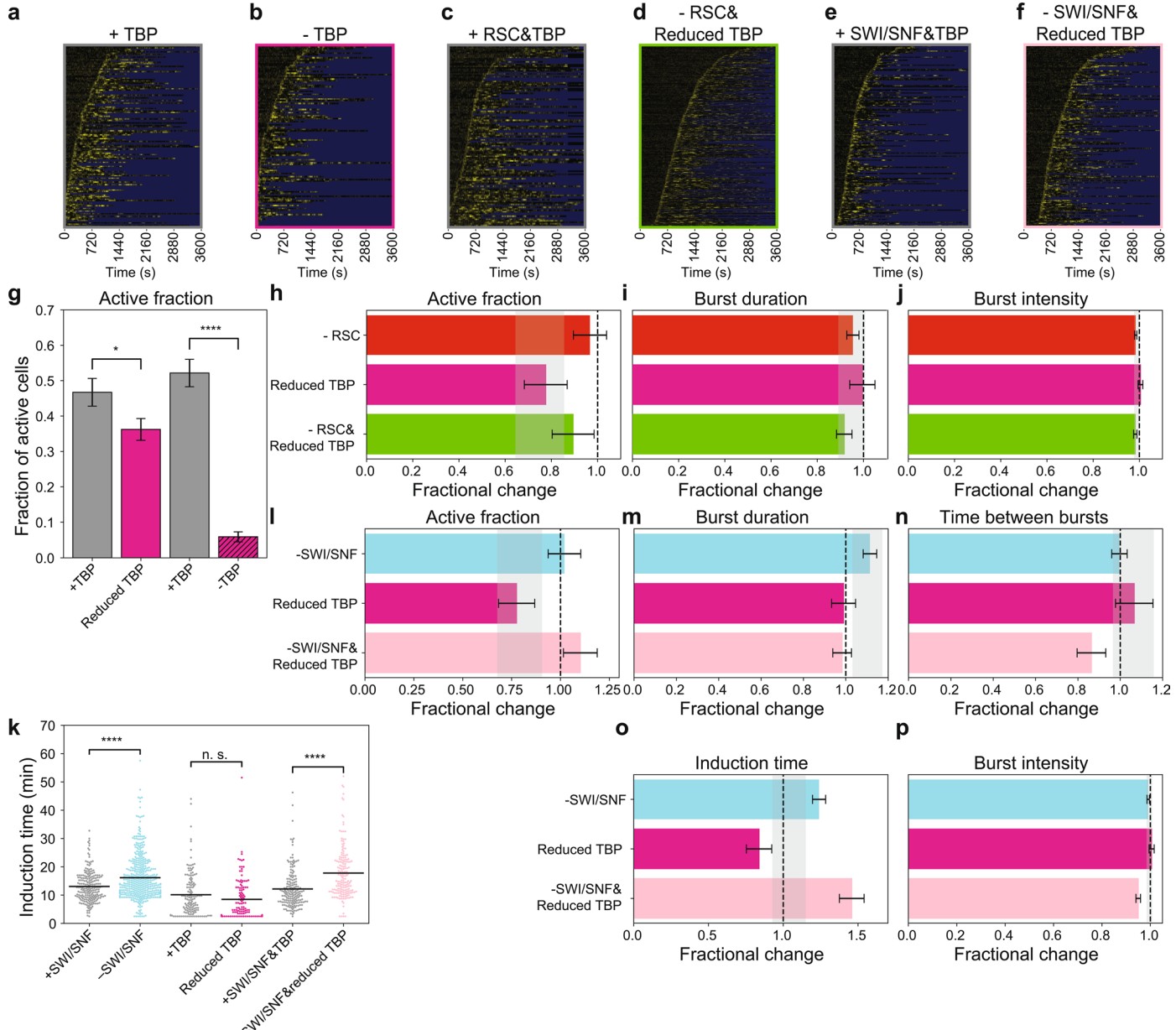

**Extended Data Fig. 5 | Transcription dynamics upon perturbed nucleosomes around the TATA, by depletion of TBP and RSC or SWI/SNF. a–f** Heatmap of TS intensity in (a) N = 139, (b) N = 120, (c) N = 122, (d) N = 340, (e) N = 179 and (f) N = 175 cells (rows) in the presence or absence of indicated factors. Yellow: TS fluorescence intensity; blue: region excluded from analysis. **g** The fraction of cells that activated during 1 hour of imaging when depleting TBP fully (-TBP, both copies tagged for depletion) or partially (reduced TBP, one copy tagged for depletion) using anchor-away. Data is the fraction based on the number of active and inactive cells +/- propagated statistical errors in these numbers. Significance determined by two-sided Fisher's exact test; *: $p < 0.05$; ****: $p < 0.00005$. $p$-values: +TBP vs partial TBP 0.0062; +/-TBP 7.7 $10^{-42}$. **h–j** The fraction of (h) cells that activated during 1 hour of imaging, (i) the burst duration and (j) the burst intensity, in -RSC&Reduced TBP were as expected based on their individual depletions. **k** SWI/SNF and simultaneous TBP and SWI/SNF depletion showed

increased *GAL10* induction time. Significance in (n) determined by two-sided bootstrap hypothesis testing[53]; n.s.: not significant; ****: $p < 0.00005$. $p$-values: +/- SWI/SNF < $10^{-15}$; +TBP vs partial TBP < $10^{-15}$; +SWI/SNF&TBP vs -SWI/SNF& partial TBP < $10^{-15}$. **l–p** SWI/SNF and simultaneous TBP and SWI/SNF depletion showed (k) increased induction time of *GAL10*. (l) The fraction of cells that activated during 1 hour of imaging, (m) the burst duration, (n) the time between bursts, (o) the induction time and (p) the burst intensity for single and double depletion of SWI/SNF and reduced TBP. (**h**)-(**j**), (**l**)-(**p**) Data for active fraction is the fractional change based on number of active and inactive cells +/- propagated statistical errors in the number of active and inactive cells. Data for other parameters are the fractional change based on mean values +/- standard deviation based on 1000 bootstrap repeats. Grey bar: expected effect based on dynamic epistasis analysis.

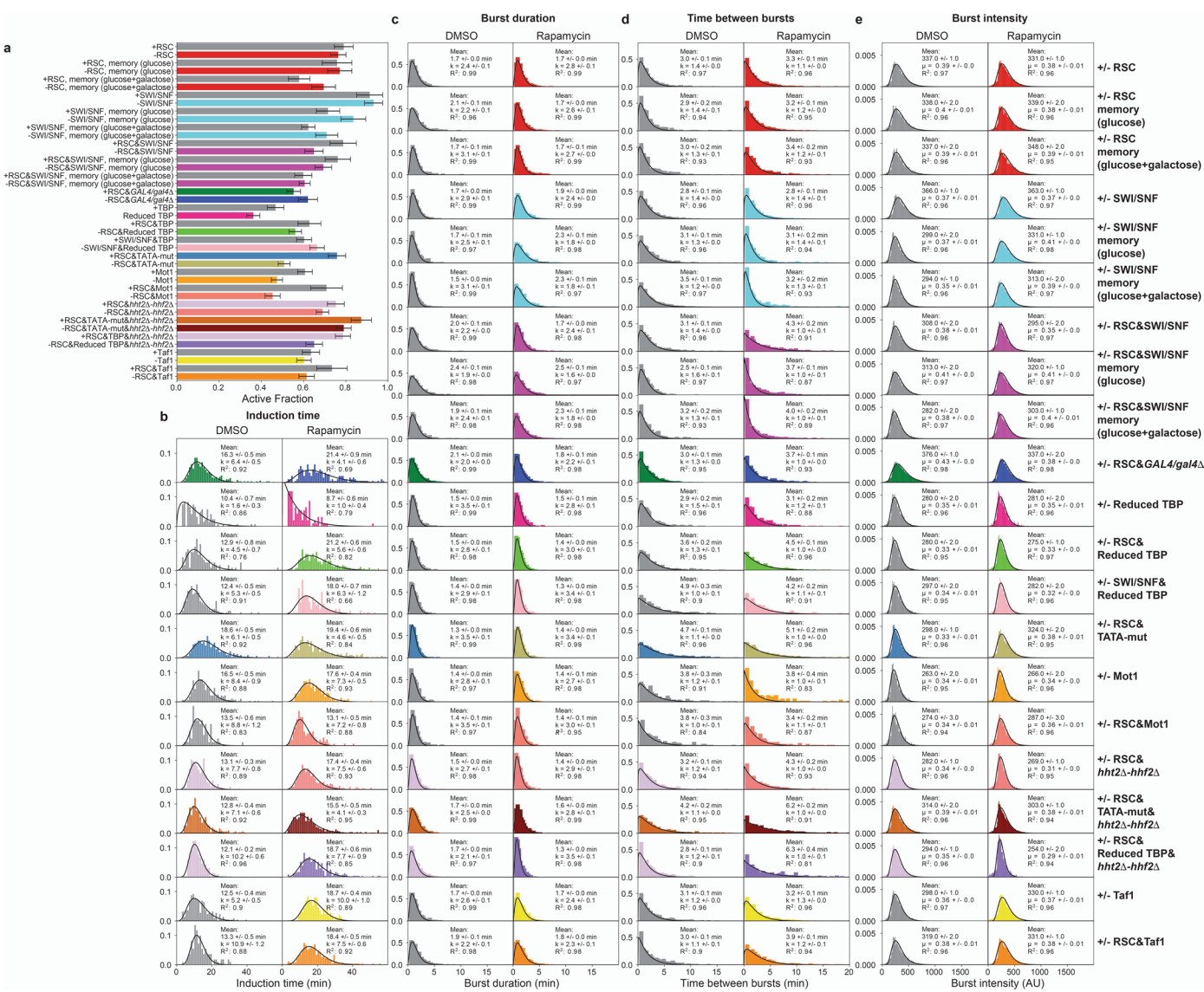

**Extended Data Fig. 6 | Distributions of the transcriptional bursting parameters for the strains used in this study. a** Active fractions for indicated strains used in this study. Error bars are propagated statistical errors in the number of active and inactive cells. **b** Distributions of the induction time for indicated strains in non-depleted (DMSO, left) or depleted (rapamycin, right) conditions. Black line: fit with Gamma distribution; inset: mean with standard deviation based on 1000 bootstrap repeats, shape parameter $k$ with error obtained from least-squares fit and $R^2$ of the fit. For the induction time distributions of +/− RSC, +/− SWI/SNF and +/− RSC&SWI/SNF in normal and memory conditions, see Extended Data Fig. 4. **c** Distributions of the burst duration for indicated strains in non-depleted (DMSO, left) or depleted

(rapamycin, right) conditions. Black line: fit with Gamma distribution; inset: mean with standard deviation based on 1000 bootstrap repeats, shape parameter $k$ with error obtained from least-squares fit and $R^2$ of the fit. **d** Distributions of the time between bursts for indicated strains in non-depleted (DMSO, left) or depleted (rapamycin, right) conditions. Black line: fit with Gamma distribution; inset: mean with standard deviation based on 1000 bootstrap repeats, shape parameter $k$ with error obtained from least-squares fit and $R^2$ of the fit. **e** Distributions of the burst intensity for indicated strains in non-depleted (DMSO, left) or depleted (rapamycin, right) conditions. Black line: fit with log-normal distribution; inset: mean with standard deviation based on 1000 bootstrap repeats, $\mu$ with error obtained from least-squares fit and $R^2$ of the fit.

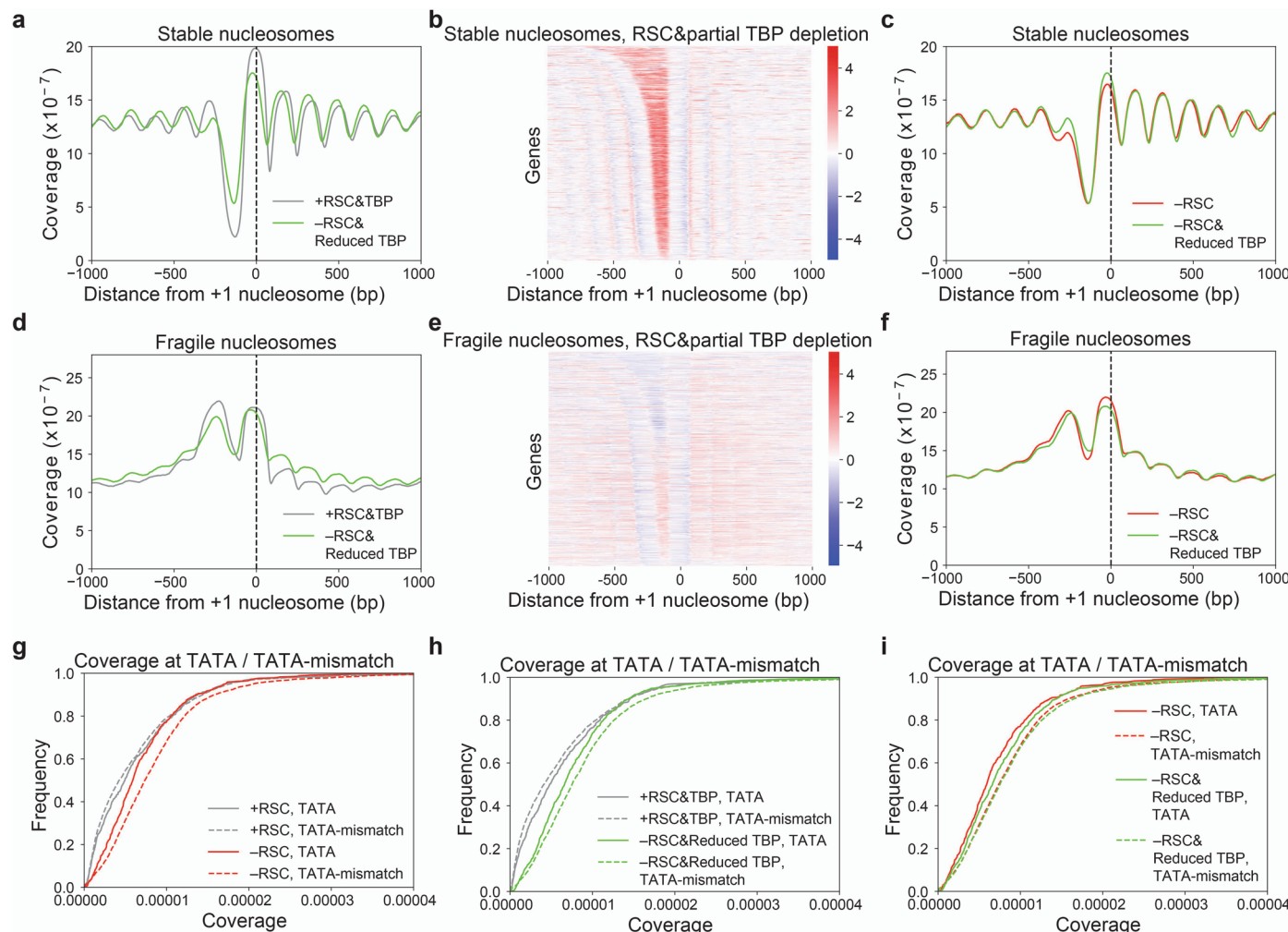

**Extended Data Fig. 7 | Genome-wide changes in nucleosome coverage upon depletion of RSC and TBP. a**, **d** Metagene MNase-seq analysis of (a) stable and (d) fragile nucleosomes upon RSC and TBP depletion, averaged over all genes as annotated by[63], aligned at the location of the +1 nucleosome (black dashed line). **b**, **e** Heatmap of the log2-fold-change of depletion/non-depleted in nucleosome coverage of (b) stable and (e) fragile nucleosomes at all genes as annotated by[63], sorted by NDR width and aligned at the location of the +1 nucleosome, upon RSC and TBP depletion. **c**, **f** Overlay of metagene MNase-seq profiles of (c) stable and (f) fragile nucleosomes upon RSC depletion and simultaneous RSC and TBP depletion, averaged over all genes as annotated by[63], aligned at the location of

the +1 nucleosome (black dashed line). **g** Cumulative distribution of coverage of stable nucleosomes in TATA or TATA-mismatch elements genome-wide upon depletion of RSC, showing a larger increase in coverage at TATA-mismatch elements than at a TATA-elements. **h** Cumulative distribution of coverage of stable nucleosomes in TATA or TATA-mismatch elements genome-wide upon simultaneous depletion of RSC and TBP. **i** Overlay of cumulative distributions of coverage of stable nucleosomes in TATA or TATA-mismatch regions genome-wide upon depletion of RSC or simultaneous depletion of RSC and TBP, showing that partial TBP depletion specifically increased the coverage at TATA-elements. Shown is one representative replicate out of two experiments for all plots.

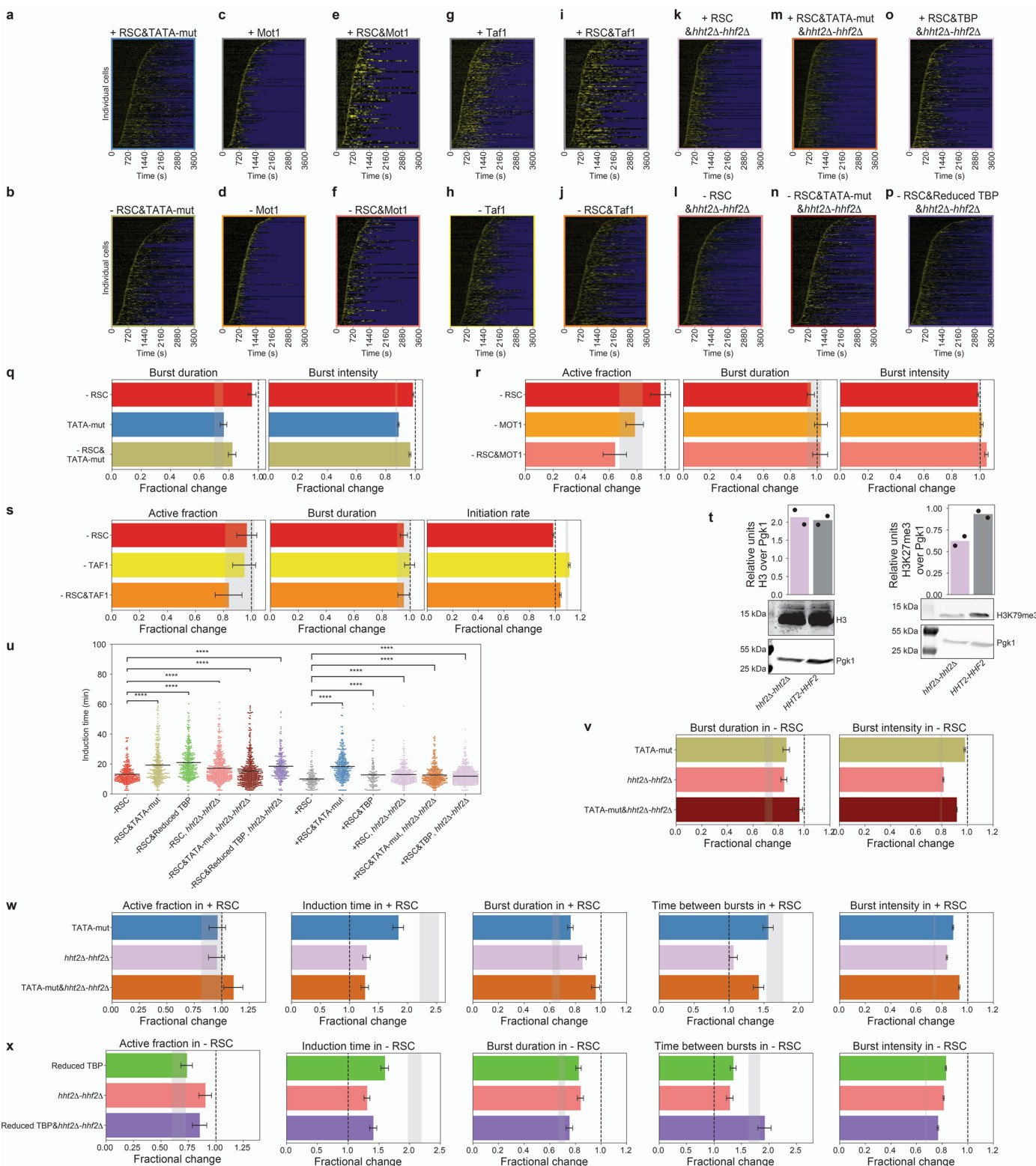

**Extended Data Fig. 8 | See next page for caption.**

**Extended Data Fig. 8 | Change in transcriptional bursting when TBP residence time is perturbed in -RSC and *hht2Δ-hhf2Δ*. a-p** Heatmap of TS intensity in (a) N = 315, (b) N = 303, (c) N = 276, (d) N = 285, (e) N = 82, (f) N = 145, (g) N = 231, (h) N = 288, (i) N = 114, (j) N = 275 cells, (k) N = 362, (l) N = 503, (m) N = 311, (n) N = 462, (o) N = 559, and (p) N = 233 (rows) in the presence or absence of indicated factors. Yellow: TS fluorescence intensity; blue: region excluded from analysis. **q** Change in indicated parameters in -RSC and/or TATA-mut. **r, s** Change in the fraction of cells that activate during 1 hour of imaging, the burst duration and the burst intensity when depleting RSC and/or (r) Mot1 or (s) Taf1. **t** Western blot analysis with H3 and H3K79me3 antibodies of *hht2Δ-hhf2Δ* and *HHT2-HHF2* cells. αH3K79me3 suggested reduced histone levels (0.67 ± 0.06 of wildtype). Shown is an example blot (N = 2) and quantification over Pgk1 levels. Black dots: individual experiments, bars: mean. **u** *GAL10* induction time in indicated strains. Both +/−RSC, induction time is increased upon TATA-mut or TBP depletion, but this is partially rescued by *hht2Δ-hhf2Δ*. Significance is by bootstrap hypothesis testing[53]; **: p < 0.005, ****: p < 0.00005. *p*-values: -RSC vs -RSC&TATA-mut <$10^{-15}$; -RSC vs -RSC&partial TBP < $10^{-15}$; -RSC vs -RSC & *hht2Δ-hhf2Δ* <$10^{-15}$; -RSC vs -RSC&TATA-mut&*hht2Δ-hhf2Δ* 0.002; -RSC vs -RSC&partial TBP&*hht2Δ-hhf2Δ* <$10^{-15}$; +RSC vs +RSC&TATA-mut <$10^{-15}$; +RSC vs +RSC&TBP < $10^{-15}$; +RSC vs +RSC&*hht2Δ-hhf2Δ* <$10^{-15}$; +RSC vs +RSC&TATA-mut&*hht2Δ-hhf2Δ* <$10^{-15}$; +RSC vs +RSC&TBP&*hht2Δ-hhf2Δ* <$10^{-15}$. **v** Change in indicated parameters in -RSC in TATA-mut and/or *hht2Δ-hhf2Δ*. Note: calculated relative to -RSC cells. **w, x** Change in indicated parameters in RSC-non-depleted cells in (w) TATA-mut or (x) partial TBP depletion and/or *hht2Δ-hhf2Δ*. Note: calculated relative to (w) +RSC cells and (x) -RSC cells. (**r**)-(**s**), (**w**)-(**x**), (**l**) Data for active fraction in is presented as the fractional change based on number of active and inactive cells +/− propagated statistical errors in these numbers. Data for the other parameters are presented as the fractional change based on mean values +/− standard deviation based on 1000 bootstrap repeats. Grey bar: expected effect based on dynamic epistasis analysis.

# Reporting Summary

## Statistics

For all statistical analyses, confirm that the following items are present in the figure legend, table legend, main text, or Methods section.

| n/a | Confirmed | |
|---|---|---|
| ☐ | ☒ | The exact sample size ($n$) for each experimental group/condition, given as a discrete number and unit of measurement |
| ☐ | ☒ | A statement on whether measurements were taken from distinct samples or whether the same sample was measured repeatedly |
| ☐ | ☒ | The statistical test(s) used AND whether they are one- or two-sided<br>*Only common tests should be described solely by name; describe more complex techniques in the Methods section.* |
| ☒ | ☐ | A description of all covariates tested |
| ☐ | ☒ | A description of any assumptions or corrections, such as tests of normality and adjustment for multiple comparisons |
| ☐ | ☒ | A full description of the statistical parameters including central tendency (e.g. means) or other basic estimates (e.g. regression coefficient) AND variation (e.g. standard deviation) or associated estimates of uncertainty (e.g. confidence intervals) |
| ☒ | ☐ | For null hypothesis testing, the test statistic (e.g. $F$, $t$, $r$) with confidence intervals, effect sizes, degrees of freedom and $P$ value noted<br>*Give P values as exact values whenever suitable.* |
| ☒ | ☐ | For Bayesian analysis, information on the choice of priors and Markov chain Monte Carlo settings |
| ☒ | ☐ | For hierarchical and complex designs, identification of the appropriate level for tests and full reporting of outcomes |
| ☒ | ☐ | Estimates of effect sizes (e.g. Cohen's $d$, Pearson's $r$), indicating how they were calculated |

*Our web collection on statistics for biologists contains articles on many of the points above.*

## Software and code

Policy information about availability of computer code

| Data collection | Micromanager software (version 1.4) for acquisition of fluorescence imaging |
|---|---|
| Data analysis | Software code for analysis of transcription dynamics microscopy data is available at https://github.com/Lenstralab/livecell. Software code for analysis of smFISH microscopy data is available at https://github.com/Lenstralab/smFISH. Software code for analysis of MNase-seq data is available at https://github.com/Lenstralab/MNase_analysis. |

For manuscripts utilizing custom algorithms or software that are central to the research but not yet described in published literature, software must be made available to editors and reviewers. We strongly encourage code deposition in a community repository (e.g. GitHub). See the Nature Portfolio guidelines for submitting code & software for further information.

## Data

Policy information about availability of data

All manuscripts must include a data availability statement. This statement should provide the following information, where applicable:
- Accession codes, unique identifiers, or web links for publicly available datasets
- A description of any restrictions on data availability
- For clinical datasets or third party data, please ensure that the statement adheres to our policy

The MNase-seq datasets generated during the current study are available in the NCBI's Gene Expression Omnibus61 through GEO Series accession number GSE190737 (https://www.ncbi.nlm.nih.gov/geo/query/acc.cgi?acc=GSE190737). For MNase-seq, reads were aligned to the reference genome SacCer3 (January 2015). MNase-seq metagene plots were generated using all verified open reading frames in the Saccharomyces Genome Database (SGD) (Cherry, J. M. et al. Saccharomyces Genome Database: The genomics resource of budding yeast. Nucleic Acids Res. 40, 700–705 (2012).)). TATA and TATA-mismatch genes were identified as 'TATA-containing' and 'TATA-less' as previously described (Rhee, H. S. & Pugh, B. F. Genome-wide structure and organization of eukaryotic pre-

# Field-specific reporting

Please select the one below that is the best fit for your research. If you are not sure, read the appropriate sections before making your selection.

☒ Life sciences ☐ Behavioural & social sciences ☐ Ecological, evolutionary & environmental sciences

For a reference copy of the document with all sections, see nature.com/documents/nr-reporting-summary-flat.pdf

# Life sciences study design

All studies must disclose on these points even when the disclosure is negative.

| | |
|---|---|
| Sample size | For live-cell fluorescence imaging, at least 100 cells were included for each condition, from at least 3 biological replicates imaged in independent experiments. This number of cells allows to robustly determine transcriptional bursting parameters, while the use of independent biological replicates eliminated bias that may have occurred because of unwanted additional random mutations in the yeast strains. For MNase-seq, two replicate experiments were performed for most conditions, as indicated in the figure legends. |
| Data exclusions | One dataset was excluded from the MNase-seq analysis based on the digestion levels as assessed on agarose gel. For live-cell imaging, cells that were segmented incorrectly and cells that contained tracking errors were excluded from analysis. In addition, we carefully checked the distributions of transcriptional bursting parameters in each biological replicate indivudally. We then found two samples (the -Taf1 dataset described in Fig. 6 and Extended Data Fig. 8 and the +Mot1&RSC dataset described in Fig. 5 and Extended Data Fig. 8) contained a fast-inducing subpopulation that arose from a single replicate experiment, suggesting that the subpopulation arose from technical rather than biological variation. The deviations of these samples may be caused by off-target mutations in these replicates, or from experimental error (for example by accidentally pre-growing cells in media with galactose instead of raffinose). When we checked how these replicates affected the analysis, we observed that exclusion of these replicates from the data resulted in the same synergies and conclusions from our dynamic epistasis analysis. Although we generally do not cherry-pick or remove outliers, we felt that the best approach is to remove these individual replicate experiments from the datasets, since the results are the same in both cases and this prevents potential overinterpretation of the subpopulations by readers. |
| Replication | all replication attempts were succesful, except the 2 experiments mentioned in the previous point. |
| Randomization | Randomization is not relevant to this study, as data was not subdivided into different experimental groups. |
| Blinding | Blinding is not relevant to this study, as data was not subdivided into different experimental groups. |

# Reporting for specific materials, systems and methods

We require information from authors about some types of materials, experimental systems and methods used in many studies. Here, indicate whether each material, system or method listed is relevant to your study. If you are not sure if a list item applies to your research, read the appropriate section before selecting a response.

## Materials & experimental systems

| n/a | Involved in the study |
|---|---|
| ☐ | ☒ Antibodies |
| ☒ | ☐ Eukaryotic cell lines |
| ☒ | ☐ Palaeontology and archaeology |
| ☒ | ☐ Animals and other organisms |
| ☒ | ☐ Human research participants |
| ☒ | ☐ Clinical data |
| ☒ | ☐ Dual use research of concern |

## Methods

| n/a | Involved in the study |
|---|---|
| ☒ | ☐ ChIP-seq |
| ☒ | ☐ Flow cytometry |
| ☒ | ☐ MRI-based neuroimaging |

# Antibodies

| | |
|---|---|
| Antibodies used | Western blot analysis was performed using antibodies against V5 (ThermoFisher R960-25, RRID: AB_2556564), Pgk1 (Invitrogen 22C5D8, RRID: AB_2532235), histone H3 (RRID:AB_2631108, a kind gift of the F.v.L. laboratory)(Frederiks, F. et al. Nonprocessive methylation by Dot1 leads to functional redundancy of histone H3K79 methylation states. Nat. Struct. Mol. Biol. 15, 550–557 (2008).) and histone H3K79me3 (RRID:AB_2631107, a kind gift of the F.v.L. laboratory)(Frederiks, F. et al. Nonprocessive methylation by Dot1 leads to functional redundancy of histone H3K79 methylation states. Nat. Struct. Mol. Biol. 15, 550–557 (2008).). Secondary antibodies used were IRDye 800CW Goat anti-Mouse IgG 925-32210 Li-COR (RRID AB_2687825), IRDye 800CW Goat anti-Rabbit IgG 926-32211 Li-COR (RRID:AB_621843) and IRDye 680RD Donkey anti-Mouse IgG 925-68072 Li-COR (RRID AB_2814912). |
| Validation | Validation information can be found at the folllowing websites for the following proteins: V5 (https://www.thermofisher.com/antibody/product/V5-Tag-Antibody-Monoclonal/R960-25) |

Pgk1 (https://www.thermofisher.com/antibody/product/PGK1-Antibody-clone-22C5D8-Monoclonal/459250)
Goat anti-Mouse (https://www.licor.com/bio/reagents/irdye-800cw-goat-anti-mouse-igg-secondary-antibody)
Goat anti-Rabbit (https://www.licor.com/bio/reagents/irdye-800cw-goat-anti-rabbit-igg-secondary-antibody)
Donkey anti-Mouse (https://www.licor.com/bio/reagents/irdye-680rd-donkey-anti-mouse-igg-secondary-antibody)

The following antibodies were kind gifts from the Fred van Leeuwen laboratory (The Netherlands Cancer Institute). Both antibodies were validated in (Frederiks, F. et al. Nonprocessive methylation by Dot1 leads to functional redundancy of histone H3K79 methylation states. Nat. Struct. Mol. Biol. 15, 550–557 (2008).):
histone H3
histione H3K79me3

