## [Peer Review File · Nature Structural & Molecular Biology]

Peer Review Information

Manuscript Title: Dynamic epistasis analysis reveals how chromatin remodeling regulates transcriptional bursting

Corresponding author name(s): Tineke (Laura) Lenstra

Reviewer Comments & Decisions:

Decision Letter, initial version:
--

Message: 28th Apr 2022

Dear Dr. Lenstra,

Thank you again for submitting your manuscript "Dynamic epistasis analysis reveals how chromatin remodeling regulates transcriptional bursting". I apologize for the delay in responding, which resulted from the difficulty in obtaining suitable referee reports, and a COVID-19 situation on our team. Nevertheless, we now have comments (below) from the 2 reviewers who evaluated your paper. In light of those reports, we remain interested in your study and would like to see your response to the comments of the referees, in the form of a revised manuscript.

You will see that reviewer #1 has some concerns regarding the quantitation of the burst parameters and normalization. Please be sure to address/respond to all concerns of the referees in full in a point-by-point response and highlight all changes in the revised manuscript text file. If you have comments that are intended for editors only, please include those in a separate cover letter.

We expect to see your revised manuscript within 6 weeks. If you cannot send it within this time, please contact us to discuss an extension; we would still consider your revision, provided that no similar work has been accepted for publication at NSMB or published elsewhere.

As you already know, we put great emphasis on ensuring that the methods and statistics reported in our papers are correct and accurate. As such, if there are any changes that should be reported, please submit an updated version of the Reporting Summary along

with your revision.

Reporting Summary:

Please note that all key data shown in the main figures as cropped gels or blots should be presented in uncropped form, with molecular weight markers. These data can be aggregated into a single supplementary figure item. While these data can be displayed in a relatively informal style, they must refer back to the relevant figures. These data should be submitted with the final revision, as source data, prior to acceptance, but you may want to start putting it together at this point.

Data availability: this journal strongly supports public availability of data. All data used in accepted papers should be available via a public data repository, or alternatively, as Supplementary Information. If data can only be shared on request, please explain why in your Data Availability Statement, and also in the correspondence with your editor. Please note that for some data types, deposition in a public repository is mandatory - more information on our data deposition policies and available repositories can be found below: <https://www.nature.com/nature-research/editorial-policies/reporting->

standards#availability-of-data

[Redacted]

Sincerely,
Sara

Sara Osman, Ph.D.
Associate Editor
Nature Structural & Molecular Biology

Referee expertise:

Referee #1: Transcription, chromatin

Referee #2: Transcription, chromatin

Reviewers' Comments:

Reviewer #1:

Remarks to the Author:

This paper cleverly takes a quantitative analysis of single-cell transcription bursting at the GAL10 promoter and applies the classic genetic approach of comparing double mutants to each of the parents, i.e. an epistasis test. In this case, the "mutants" are anchor-away depletions, and the "phenotypes" are the transcription burst parameters. Starting with a simple model where the two factors act independently, stronger than expected effects can suggest cooperation/redundancy and weaker effects (i.e. suppression) can suggest opposing effects. Integrating these experiments with analysis of nucleosomes at GAL10, a model for how various factors regulate bursting is proposed. It's a nice study that is reasonably convincing and well written. I have some questions about quantitation and interpretations, and some things that are stated as mechanistic conclusions are really models or correlations.

1. My main concern is the quantitation of the burst parameters. Throughout the paper, parameters are quantitated as fractional changes in the mean of the population. While this simplifies the analysis, there are two problems with this approach.

A. Using a simple mean works if all the cells behave as a single population type with a normal or exponential distribution, but the big advantage of single cell measurements is you can see if this is true. The gamma function analysis suggests it might not be. Looking at the violin plots, sometimes the average changes because entire population shifts, but in others the effect is mostly because the late-inducing cells stretch out longer (comparing the mode to the average). It also looks like there are two populations of cells in some cases, most notably in Fig 6B and ED Fig 7H. The TAF1 depletion shows a subpopulation of cells inducing faster than normal within the first few minutes, then there is a gap, and a second population of cells induces at 10 minutes or longer. By using the average, the advantage of single cell measurements can be lost, affecting the model. Maybe using the average is OK for a quick first pass, but the authors might want to consider whether more sophisticated analyses can provide better insight.

B. The second questionable thing is normalization of all the pre-depletion samples to 1.0. Theoretically these should all be the same, essentially the "wild-type" of the epistasis analysis. But the violin plots show that the different AID-tagged strains are quite different, often as much or more than the +/- depletion differences. For example, in ED Fig 3 there's a clear difference between panel b (+SWI/SNF) and d (+RSC&SWI/SNF). ED Fig 5n shows the TBP-AID strain induces significantly faster, even before depletion. Fig 5G shows a big difference between the +RSC and the +Mot1 controls. Does this indicate the AID tags are already affecting function? Or are these strains non-isogenic? Or is this variability because the different samples were analyzed on different days? The normalized fractional bar graphs hide all these differences. In only a few cases are the corresponding violin plots shown, so adding a supplemental file with all of these would be helpful.

3. Line 110. Why is there less fragile nucleosome upon RSC depletion? Shouldn't this condition stabilize it? Could the "fragile nucleosome" actually be other TFs or GTFs that are released at low MNase?

4. Line 134-136, 189-192: These sentences conclude that the RSC/SNF2 effects on fragile nucleosomes are responsible for the differences in burst properties. The data shows a correlation, but doesn't prove causation. RSC could have multiple functions.

5. Fig 3: Has it been shown that GAL4/gal4delta really has lower Gal4 protein levels, to rule out post-transcriptional compensation at the mRNA or protein level?

6. Line 291/Fig 4. The text says that anchor away of one TBP allele reduces the on rate of TBP at the Gal locus. Is there data that this is true, or should this be stated as an assumption? The extended figure shows reduction of nuclear TBP signal, but doesn't assay binding. Line 298 says there's only a "modest effect" of TBP depletion, but the data actually shows a paradoxical decrease in induction time. This doesn't really fit the model.

7. Fig 5., Line 338-348. The drop in active fraction in the TATA mutant plus RSC depletion is said to be surprising evidence for two populations. But from the other figures it looks like pretty much any of the other depletions combined with RSC depletion give the same effect. The paper describes a TBP-nucleosome competition as if proven, but with no single cell data on nucleosome position it should be made clear this is speculation. You could test whether partial histone depletion rescues the silent population (similar to the Mot1 codepletion). Histone depletion by anchor away would probably be difficult, but you can knock out one of the two histone gene copies, which has long been known to change transcription. Assaying histone dosage changes in this single cell bursting assay would more directly test the paper's model about nucleosome competition. There's also a 2016 eLife paper from Erin O'Shea (PMC5094857) that contains nucleosome mapping data that may be relevant to this model.

8. Line 372. What's the evidence that Taf1 binds after TBP? My understanding is the usual model for TFIID is that this TBP-TAF complex functions as a single complex, while more recent models based on cryoEM structures say the TAFs actually bind first and recruit TBP.

9. Lines 437 and 445, summarizing data comparing promoters with consensus versus non-consensus TATA sequences, says the differences reflect TBP residence time. Is there data measuring residence time at these promoters, or is this another assumption?

10. Line 446. If TBP can stably bind the nucleosome as stated, how would longer binding time lead to nucleosome eviction or sliding? The TBP-nucleosome complex is either stable or unstable, so it's not clear what thermodynamic model the authors have in mind.

Minor comments:

ED Fig 1 heat maps. The red/blue scale is presumably a ratio. Is it depleted/undepleted or undepleted/depleted? I'm confused, because for the stable nucleosomes the largest change is in the NDR itself.

Line 1052 (ED Fig4). "A k-value significantly different from 1 indicates a single rate-limiting step". Shouldn't this be "not significantly different"?

ED Fig 4. The labeling for this experiment is confusing because "glucose repression"

should cause a slower response, but here "repression" induces faster. The text explains this is actually a "memory" experiment, so I suggest re-labeling the three conditions as "non-memory", "memory (glucose)", and "memory (glucose+galactose)" to make this clear.

Reference 26 is incomplete - no page numbers or doi

Reviewer #2:

Remarks to the Author:

This is a very interesting paper that significantly contributes to the field of single molecule studies and quantification of transcription. This paper provides mechanistic insights on interaction of the transcription factors and chromatin remodelers. The authors address and quantify important issues – how do chromatin remodelers affect chromatin structure and, specifically, promoter-specific nucleosomal landscape, how the chromatin remodelers achieve the specificity of their action on specific promoters, how different types of chromatin remodelers interact on a same promoter, and how the transient remodeling by multiple remodelers affects gene activation and parameters of transcription. Those questions are of a significant and immediate general interest. Currently we possess only a general understanding of the dynamics of chromatin remodelers at promoters. We do not know how different types of remodelers interact at specific promoters. We may expect multiple mechanisms of the remodeler activity at individual promoters. The new and original impact of this paper is the interplay of the remodelers and transcription factors revealed at nucleosome level. This paper will influence the thinking in the field by demonstrating a novel mechanism of nucleosomal specificity of different remodelers at the same promoter. This paper deserves publication in Nature Structural and Molecular Biology.

The authors demonstrated that transcription of the yeast GAL10 depends only on two chromatin remodeler complexes – RSC and SWI/SNF. They explored in detail the activity of RSC and SWI/SNF on three nucleosomes of GAL10 promoter and then investigated the interactions of RSC and SWI/SNF with transcription factors. They observed the nucleosomal landscape by MNase-seq and observed dynamic effects of single and double depletions of remodelers and transcription factors in live cells with a stem-loop approach. Stem-loop data were corroborated by smFISH.

First outstanding and novel conclusion is the discrete action of different chromatin remodelers on the nucleosomes of GAL10 promoter, rather than competition for the same nucleosomes. RSC regulates fragile nucleosome over the GAL10 UAS, and SWI/SNF mostly affect stable nucleosome at TATA-box of GAL10. They overlap in functions over the TSS nucleosome.

The second outstanding and novel conclusion is a nucleosome-specific dual role of RSC in GAL10 transcription: RSC dynamically regulates transcription bursts via fragile UAS nucleosome; RSC cooperates with SWI/SNF in mobilizing TSS and TATA nucleosomes and establishing transcription competency at the step of initial GAL10 induction.

The third outstanding and novel conclusion provides the insight into the mechanism of interaction of RSC with transcription activator Gal4. They interact in a different way at the initial step of induction, and at later stages, when transcriptional competence is established. The authors observed dependence of burst frequency on synergistic action of RSC and transcriptional activator GAL4; however, RSC and Gal4 appear to act independently at the step of initial GAL10 induction. Thus, Gal4 acts as a pioneer factor. The fourth outstanding and novel conclusion provides the insight into the role of the

optimized residence time of transcription factors TBP and Taf1. Depletion of MOT1, which at this promoter is involved in removal of TBP, compensates for RSC depletion. The authors suggest that extended residence time of TBP may lead to nucleosome eviction and compensate for the lack of remodeling activity. The authors also observed destabilizing effect of RSC on Taf1 binding. Specifically, depletion of Taf1 compensates for the function of RSC at the step of initial GAL10 induction.

In sum, the authors uncover a novel mechanism of regulation dependent on differential remodeling of individual nucleosomes. Fragile nucleosome bound to UAS is dynamically displaced by either Gal4 or RSC, leading to subsequent bursts. Bursts require specific positioning of TATA and TSS nucleosomes. This positioning is established and maintained by SWI/SNF, and TBP, and RSC, which plays a role on TSS nucleosome, different from that on UAS nucleosome.

This paper provides strong evidence for its conclusions. The authors applied a variety of methods to support their data. The data are adequately quantified. This paper is a delight to read: anything you may think to suggest, the authors already did. This is a high-quality careful study – the authors even checked the cell-specific effects of remodeler depletion. It would be very interesting to dissect the effect of two specific complexes, RSC1 and RSC2, on the transcription of GAL10, and this may be a subject of a new study, or, alternatively, if authors have any data or opinions on this issue, comments could be added to the Discussion of this paper.

I recommend this paper for publication. Its length is appropriate to describe the results.

Author Rebuttal to Initial comments

Reviewers' Comments:

Reviewer #1:

Remarks to the Author:

This paper cleverly takes a quantitative analysis of single-cell transcription bursting at the GAL10 promoter and applies the classic genetic approach of comparing double mutants to each of the parents, i.e. an epistasis test. In this case, the “mutants” are anchor-away depletions, and the “phenotypes” are the transcription burst parameters. Starting with a simple model where the two factors act independently, stronger than expected effects can suggest cooperation/redundancy and weaker effects (i.e. suppression) can suggest opposing effects. Integrating these experiments with analysis of nucleosomes at *GAL10*, a model for how various factors regulate bursting is proposed. It's a nice study that is reasonably convincing and well written. I have some questions about quantitation and interpretations, and some things that are stated as mechanistic conclusions are really models or correlations.

We thank the reviewer for his/her positive evaluation of our approach. We appreciate the constructive comments and proposed experiments, which greatly improved the manuscript. Our point-by-point responses are in blue below.

1. My main concern is the quantitation of the burst parameters. Throughout the paper, parameters are quantitated as fractional changes in the mean of the population. While this simplifies the analysis, there are two problems with this approach.

A. Using a simple mean works if all the cells behave as a single population type with a normal or exponential distribution, but the big advantage of single cell measurements is you can see if this is true. The gamma function analysis suggests it might not be. Looking at the violin plots, sometimes the average changes because entire population shifts, but in others the effect is mostly because the late-inducing cells stretch out longer (comparing the mode to the average). It also looks like there are two populations of cells in some cases, most notably in Fig 6B and ED Fig 7H. The TAF1 depletion shows a subpopulation of cells inducing faster than normal within the first few minutes, then there is a gap, and a second population of cells induces at 10 minutes or longer. By using the average, the advantage of single cell measurements can be lost, affecting the model. Maybe using the average is OK for a quick first pass, but the authors might want to consider whether more sophisticated analyses can provide better insight.

We agree with the reviewer that comparison of the mean of two distributions is mostly useful when the shape of the two distributions is similar. To allow the reader to evaluate changes in the shape of the distribution, we have now included the full distributions of all kinetic parameters of transcriptional bursting in Extended Data Fig. 8.

To determine whether there are subpopulations in the datasets exhibiting different behavior, we tested whether the shapes of all distributions were well-described by a theoretical distribution. For the induction time, time between bursts and burst duration, we fit a Gamma distribution and for the burst intensity we fit a log-normal distribution. Theoretically, the burst duration should be described by the sum of a deterministic time (the elongation time for *GAL10*) and an exponential distribution describing the initiation kinetics, but we approximated this with a Gamma distribution for simplicity. The goodness of the fit was determined by calculating the R^2 and the residuals between the data and the fit.

For the time between bursts, Gamma fits consistently showed k between 1 and 1.6 with $R^2 > 0.8$ (Extended Data Fig. 8), indicating these distributions are approximately described by single populations with an exponential distribution. The fits to the burst duration and burst intensity distributions showed high R^2 values (>0.97 and >0.94 respectively, Extended Data Fig. 8). However, for the induction time distributions, we found 6 experiments with R^2 values below 0.8, indicating that these distributions are not well-described by a Gamma distribution. This could be

caused by the presence of multiple populations. We carefully checked these experiments and noted that two samples (the -Taf1 dataset described in Fig. 6 and Extended Data Fig. 7, as noted by the reviewer, and the +Mot1&RSC dataset described in Fig. 5 and Extended Data Fig. 7) contained a fast-inducing subpopulation that arose from a single replicate experiment, suggesting that the subpopulation arose from technical rather than biological variation. The deviations of these samples may be caused by off-target mutations in these replicates, or from experimental error (for example by accidentally pre-growing cells in media with galactose instead of raffinose). When we checked how these replicates affected the analysis, we observed that exclusion of these replicates from the data resulted in the same synergies and conclusions from our dynamic epistasis analysis (Figure R1 and R2). Although we generally do not cherry-pick or remove outliers, we felt that the best approach is to remove these individual replicate experiments from the datasets, since the results are the same in both cases and this prevents potential overinterpretation of the subpopulations by readers. If, however, the reviewers or editor find this unacceptable, we are happy to keep it in the manuscript and comment on the technical nature of the subpopulations in the manuscript instead. To help the editor and reviewer judge this choice, we have included the full distributions of the induction times before and after exclusion of this single replicate experiment, the distributions of each replicate experiment separately and the effect the replicates have on the epistasis analysis (Figure R1 and R2).

The remaining 4 experiments with low R^2 values of the induction time fit did not show signs of subpopulations. Three of these describe experiments where TBP was tagged for depletion (Extended Data Fig. 8) and these distributions also deviated substantially from other non-depleted datasets, suggesting that the AA-tag may partially interfere with TBP function. This may also explain the faster induction of TBP upon depletion (see next point). For the last experiment, *GAL4/gal4Δ* upon RSC depletion, the low R^2 value appeared to be caused by data sparsity, even though we included 181 cells. This data sparsity also appeared to cause inconsistencies in the k values of the Gamma fits to the induction times, preventing proper interpretation. In addition, the inconsistent k values could arise because the Gamma distribution assumes different rate-limiting steps with equal rates, which may not be valid in all conditions. We now describe this subpopulation analysis in the methods, comment on the TBP tagging artifact in the main text and show the individual distributions and fits in Extended Data Fig. 8.

Overall, our new analysis shows no signs of additional subpopulations, suggesting that the mean represents the entire population. Moreover, it is important to note that the generalized bootstrap mean that we use in our analysis is valid for all distribution shapes, justifying our analysis using the fractional change of the mean to describe the changes in transcriptional bursting in the entire population.

Figure R1: induction time of the +Mot1&RSC dataset. Distribution of induction time before (left) and after (middle) exclusion of the fast-inducing subpopulation with corresponding fits to a Gamma distribution. Residuals of the fit are shown below the distributions. Corresponding synergy plots are in the bottom panels. The results from the three independent biological replicates are shown (right). Replicate 1 shows much faster induction kinetics and thus is the cause of the fast-inducing subpopulation in the left panel.

Figure R2: induction time of the -Taf1 dataset. Distribution of induction time before (left) and after (middle) exclusion of the fast-inducing subpopulation with corresponding fits to a Gamma distribution. Residuals of the fit are shown below the distributions. Corresponding synergy plots are in the bottom panels. The results from the three independent biological replicates are shown (right). Replicate 3 shows much faster induction kinetics and thus is the cause of the fast-inducing subpopulation in the left panel.

B. The second questionable thing is normalization of all the pre-depletion samples to 1.0. Theoretically these should all be the same, essentially the “wild-type” of the epistasis analysis. But the violin plots show that the different AID-tagged strains are quite different, often as much or more than the +/- depletion differences. For example, in ED Fig 3 there’s a clear difference between panel b (+SWI/SNF) and d (+RSC&SWI/SNF). ED Fig 5n shows the TBP-AID strain induces significantly faster, even before depletion. Fig 5G shows a big difference between the +RSC and the +Mot1 controls. Does this indicate the AID tags are already affecting function? Or are these strains non-isogenic? Or is this variability because the different samples were analyzed on different days? The normalized fractional bar graphs hide all these differences. In only a few cases are the corresponding violin plots shown, so adding a supplemental file with all of these would be helpful.

We agree with the reviewer that there is a significant amount of variation between the different non-depleted samples. As indicated by the reviewer, tagging artifacts, off-target mutations, or experiment-to-experiment variation may contribute to the observed variability. We aimed to minimize the effects of off-target mutants by including at least two, and often three, independently constructed biological replicates for each experiment. For TBP, there are three indications that addition of the anchor-away tag may partially affect its function. First, the Gamma fits to the induction time distributions show low R^2 , as described in the analysis above (Extended Data Fig. 8). Second, the shapes of the distributions deviate substantially from other non-depleted datasets. Third, cells with partial depletion of TBP showed faster induction than cells with non-depleted TBP (Figure 4 and Extended Data Figure 5), which is highly unexpected for an essential protein such as TBP. We now discuss these potential sources of variability and the potential tagging artifact of TBP in the main text. We note that our “normalization” approach circumvents this variability and allows to analyze the specific effects of single and double perturbations. Nevertheless, we agree that the unnormalized data should be available for the reader, and we have now included the full distributions of each parameter in Extended Data Fig. 8.

3. Line 110. Why is there less fragile nucleosome upon RSC depletion? Shouldn't this condition stabilize it? Could the “fragile nucleosome” actually be other TFs or GTFs that are released at low MNase?

Reads in the low MNase samples could in principle reflect other complexes. To enrich for nucleosomal particles, our analysis only included reads with a footprint in the nucleosomal range (filtered for 95-225 bp). In the field, there has been an active discussion on the identity of these particles^{1,2}. In 2019, the Henikoff lab used CUT&RUN to show that these fragile particles represent RSC-bound, partially unwrapped nucleosomal intermediates³. We rephrased the sentence in the main text to make this clearer: “We also observe a slightly lower coverage of fragile nucleosomes in promoter regions (Extended Data Fig. 1,f,g), in agreement with the finding that fragile nucleosome represent RSC-bound, partially unwrapped nucleosomal intermediates⁴.”

4. Line 134-136, 189-192: These sentences conclude that the RSC/SNF2 effects on fragile nucleosomes are responsible for the differences in burst properties. The data shows a correlation, but doesn't prove causation. RSC could have multiple functions.

We thank the reviewer for pointing this out. We have reformulated these sentences to describe the correlation without concluding causation: “Remodeling of the fragile *GAL10* promoter nucleosomes at the UASs and the TSS by RSC is thus correlated with changes in the induction

time as well as the start of each burst of *GAL10* transcription.” and “these results showed that remodeling of the fragile nucleosomes at the UASs by RSC, and nucleosome displacement around the TATA element and the TSS by RSC and SWI/SNF, are associated with synergistic changes in the induction time, time between bursts and burst size”

5. Fig 3: Has it been shown that *GAL4/gal4delta* really has lower Gal4 protein levels, to rule out post-transcriptional compensation at the mRNA or protein level?

To verify that Gal4 protein was reduced in the *GAL4/gal4Δ* strain, we added a 3V5 tag to Gal4 in the *GAL4/GAL4* and the *GAL4/gal4Δ* strains, and quantified its levels by Western Blot. We found that Gal4 was expressed at 0.5 ± 0.1 of wildtype levels, as expected. The results are now included in Extended Data Fig. 3 and described in the main text: “Western blot analysis showed that deletion of one of the two *GAL4* gene copies resulted in a 2-fold reduction of Gal4 levels (Extended Data Fig. 3m)”

6. Line 291/Fig 4. The text says that anchor away of one TBP allele reduces the on rate of TBP at the Gal locus. Is there data that this is true, or should this be stated as an assumption? The extended figure shows reduction of nuclear TBP signal, but doesn't assay binding. Line 298 says there's only a “modest effect” of TBP depletion, but the data actually shows a paradoxical decrease in induction time. This doesn't really fit the model.

We have not directly measured the on-rate of TBP at the *GAL10* TATA element. We therefore changed the phrasing to “which is expected to reduce the on-rate”. We now also comment in the main text on the unexpected decrease in induction time after partial TBP depletion that may arise from tagging artifacts: “Surprisingly, partial TBP depletion resulted in slightly faster induction than no depletion (Fig. 4b-c, Extended Data Fig. 5a-b). For this condition, the shape of the distribution of induction times in both depleted and non-depleted conditions deviated substantially from the other non-depleted conditions and showed low correlations to the theoretical Gamma distribution (Extended Data Fig. 8, Methods “Testing for subpopulations”), suggesting that fusion of TBP with the anchor-away tag may partially affect its function.”

7. Fig 5., Line 338-348. The drop in active fraction in the TATA mutant plus RSC depletion is said to be surprising evidence for two populations. But from the other figures it looks like pretty much any of the other depletions combined with RSC depletion give the same effect. The paper describes a TBP-nucleosome competition as if proven, but with no single cell data on nucleosome position it should be made clear this is speculation. You could test whether partial histone depletion rescues the silent population (similar to the Mot1 codepletion). Histone depletion by anchor away would probably be difficult, but you can knock out one of the two histone gene

copies, which has long been known to change transcription. Assaying histone dosage changes in this single cell bursting assay would more directly test the paper's model about nucleosome competition. There's also a 2016 eLife paper from Erin O'Shea (PMC5094857) that contains nucleosome mapping data that may be relevant to this model.

We thank the reviewer for the suggestion to test the nucleosome competition model further using histone deletion mutants. If TBP competes with nucleosomes, it is expected that impaired TBP binding due to TATA mutation or reduced TBP levels is rescued if histones are less abundant. To test this, we have analyzed how deletion one of the two H3 and H4 gene copies, *hht2Δ-hhf2Δ*, affects bursting in RSC-depleted cells with a TATA mutation or with partial TBP depletion. Although a Western blot with the H3 antibody did not detect a change in histone levels in *hht2Δ-hhf2Δ* compared to wildtype (1.0 ± 0.1), the H3K79me3 antibody detected a reduction of 0.33 ± 0.06 fold upon partial histone deletion (Extended Data Figure 7). Because the H3K79me3 modification is present on 90% of the nucleosomes, the measured reduction with this antibody suggested these strains contain reduced histone levels. Measurements of transcriptional bursting in these strains revealed that partial histone deletion affected several transcriptional bursting parameters, such as the induction time and the time between bursts (Figure 5 and Extended Data Figure 7). As hypothesized, the inactive population that was present in the TATA mutant upon depletion of RSC, was absent when histones were also deleted, as demonstrated by a complete rescue of the active fraction. The ability of TBP to compete with nucleosomes is thus restored if nucleosomes are less abundant. In addition, both in the absence and in the presence of RSC, the effect of TATA mutation on several other bursting parameters was partially or fully rescued by partial histone deletion (Figure 5 and Extended Data Figure 7). To verify this result further, we combined partial TBP depletion with partial histone deletion in the absence of RSC and found a similar rescue of the bursting phenotypes of the partial TBP depletion (Extended Data Figure 7). The only bursting parameter that did not show a rescue by partial histone deletion was the time between bursts, in line with our model that the TBP-nucleosome competition does not regulate the time between bursts. Since histone gene deletion is not a conditional perturbation, it is important to note that indirect effects may also contribute to the observed rescue of the transcription effects. Nevertheless, these results agree with the model that TBP competes with nucleosomes to regulate transcription. We have included these results in the manuscript, Figure 5 and Extended Data Figure 7. We have also reformulated our conclusion on the TBP-nucleosome competition to indicate that this is a model and not proven.

In addition, we agree with the reviewer that in some of the other conditions where RSC is depleted in combination with another factor, there is also a drop in active fraction, but except for the TATA mutation combined with RSC depletion, none of these are smaller than the expected

range based on the individual perturbations. We have now described this more explicitly in the text: "as evidenced by a much smaller than expected fraction of cells that activated transcription during our 1-hour imaging experiment".

As suggested by the reviewer, we used the approach developed in the 2016 eLife paper from the lab of Erin O'Shea to extract base-pair resolution nucleosomes positions in our MNase data⁵. The results are indicated below (Figure R3), where each detected nucleosome is indicated as a circle of which the intensity represents the frequency of a nucleosome occurring at that position in the population. We noted that the detection of nucleosomes is highly dependent on the number of reads per experiment (Figure R3, right) and therefore cause a substantial amount of variability. In one experiment (+RSC), the algorithm could not reliably detect any nucleosome positions due to the low low-read number of this sample. Overall, the analysis agreed with our model and revealed additional nucleosomes around the TATA element (brown line) upon RSC, RSC&TBP, SWI-SNF and RSC&SWI/SNF depletion. However, the inability to normalize/correct for the number of reads argued against including this data in the manuscript.

Figure R3: analysis of MNase data using methodology developed by the lab of Erin O'Shea⁵. Left: observed nucleosome positions around the GAL10 promoter. Circles indicate observed nucleosome positions, shading indicates frequency of occupancy of this position in the given dataset. Green region: GAL10 UASs, brown region: TATA, dashed line: TSS. Right: number of reads in the nucleosomal range (95-225 bp) for each dataset.

8. Line 372. What's the evidence that Taf1 binds after TBP? My understanding is the usual model for TFIID is that this TBP-TAF complex functions as a single complex, while more recent models based on cryoEM structures say the TAFs actually bind first and recruit TBP.

We have removed this line to correct this.

9. Lines 437 and 445, summarizing data comparing promoters with consensus versus non-consensus TATA sequences, says the differences reflect TBP residence time. Is there data measuring residence time at these promoters, or is this another assumption?

We agree this has not been measured, and have reformulated these sentences to: " This role for TBP in nucleosome positioning appears more important for genes with a canonical TATA box, possibly by supporting longer TBP residence times" and " the ability of TBP to compete with nucleosomes may depend on the residence time of TBP at the TATA box"

10. Line 446. If TBP can stably bind the nucleosome as stated, how would longer binding time lead to nucleosome eviction or sliding? The TBP-nucleosome complex is either stable or unstable, so it's not clear what thermodynamic model the authors have in mind.

We have adjusted the text in the discussion to make our model more clear. We now describe two alternative mechanisms for TBP-nucleosome competition. "We envision a passive competition mechanism where binding of TBP may perturb the stability of the nucleosome by recruiting TFIIA and other PIC components to partially unwrapped nucleosomes intermediates that arise from spontaneous nucleosome breathing. If the residence time of TBP is long enough, successive binding of multiple PIC components may eventually compete away the nucleosome, resulting in a nucleosome-free TATA box needed for recruitment of the remaining PIC components. Alternatively, the bending of the DNA that is introduced by binding of TBP may change the nucleosome positioning energy landscape. Longer TBP binding may increase the probability of downstream nucleosome movement to energetically more favorable sites. "

Minor comments:

ED Fig 1 heat maps. The red/blue scale is presumably a ratio. Is it depleted/undepleted or undepleted/depleted? I'm confused, because for the stable nucleosomes the largest change is in the NDR itself.

We adjusted to text to make clear the fold change is depleted/non-depleted.

Line 1052 (ED Fig4). “A k-value significantly different from 1 indicates a single rate-limiting step”. Shouldn’t this be “not significantly different”?

We thank the reviewer from pointing out this mistake and have corrected the text according to the suggestion.

ED Fig 4. The labeling for this experiment is confusing because “glucose repression” should cause a slower response, but here “repression” induces faster. The text explains this is actually a “memory” experiment, so I suggest re-labeling the three conditions as “non-memory”, “memory (glucose)”, and “memory (glucose+galactose)” to make this clear.

We appreciate this suggestion, and have adjusted the figure accordingly.

Reference 26 is incomplete - no page numbers or doi

We thank the reviewer for pointing out this oversight and have updated the reference.

Reviewer #2:

Remarks to the Author:

This is a very interesting paper that significantly contributes to the field of single molecule studies and quantification of transcription. This paper provides mechanistic insights on interaction of the transcription factors and chromatin remodelers. The authors address and quantify important issues – how do chromatin remodelers affect chromatin structure and, specifically, promoter-specific nucleosomal landscape, how the chromatin remodelers achieve the specificity of their action on specific promoters, how different types of chromatin remodelers interact on a same promoter, and how the transient remodeling by multiple remodelers affects gene activation and parameters of transcription. Those questions are of a significant and immediate general interest. Currently we possess only a general understanding of the dynamics of chromatin remodelers at promoters. We do not know how different types of remodelers interact at specific promoters. We may expect multiple mechanisms of the remodeler activity at individual promoters. The new and original impact of this paper is the interplay of the remodelers and transcription factors revealed at nucleosome level. This paper will influence the thinking in the field by demonstrating a novel mechanism of nucleosomal specificity of different remodelers at the same promoter. This paper deserves publication in Nature Structural and Molecular Biology. The authors demonstrated that transcription of the yeast GAL10 depends only on two chromatin remodeler complexes – RSC and SWI/SNF. They explored in detail the activity of RSC and SWI/SNF

on three nucleosomes of GAL10 promoter and then investigated the interactions of RSC and SWI/SNF with transcription factors. They observed the nucleosomal landscape by MNase-seq and observed dynamic effects of single and double depletions of remodelers and transcription factors in live cells with a stem-loop approach. Stem-loop data were corroborated by smFISH.

First outstanding and novel conclusion is the discrete action of different chromatin remodelers on the nucleosomes of GAL10 promoter, rather than competition for the same nucleosomes. RSC regulates fragile nucleosome over the GAL10 UAS, and SWI/SNF mostly affect stable nucleosome at TATA-box of GAL10. They overlap in functions over the TSS nucleosome. The second outstanding and novel conclusion is a nucleosome-specific dual role of RSC in GAL10 transcription: RSC dynamically regulates transcription bursts via fragile UAS nucleosome; RSC cooperates with SWI/SNF in mobilizing TSS and TATA nucleosomes and establishing transcription competency at the step of initial GAL10 induction.

The third outstanding and novel conclusion provides the insight into the mechanism of interaction of RSC with transcription activator Gal4. They interact in a different way at the initial step of induction, and at later stages, when transcriptional competence is established. The authors observed dependence of burst frequency on synergistic action of RSC and transcriptional activator GAL4; however, RSC and Gal4 appear to act independently at the step of initial GAL10 induction. Thus, Gal4 acts as a pioneer factor.

The fourth outstanding and novel conclusion provides the insight into the role of the optimized residence time of transcription factors TBP and Taf1. Depletion of MOT1, which at this promoter is involved in removal of TBP, compensates for RSC depletion. The authors suggest that extended residence time of TBP may lead to nucleosome eviction and compensate for the lack of remodeling activity. The authors also observed destabilizing effect of RSC on Taf1 binding. Specifically, depletion of Taf1 compensates for the function of RSC at the step of initial GAL10 induction.

In sum, the authors uncover a novel mechanism of regulation dependent on differential remodeling of individual nucleosomes. Fragile nucleosome bound to UAS is dynamically displaced by either Gal4 or RSC, leading to subsequent bursts. Bursts require specific positioning of TATA and TSS nucleosomes. This positioning is established and maintained by SWI/SNF, and TBP, and RSC, which plays a role on TSS nucleosome, different from that on UAS nucleosome. This paper provides strong evidence for its conclusions. The authors applied a variety of methods to support their data. The data are adequately quantified. This paper is a delight to read: anything you may think to suggest, the authors already did. This is a high-quality careful study – the authors even checked the cell-specific effects of remodeler depletion. It would be very interesting to dissect the effect of two specific complexes, RSC1 and RSC2, on the transcription of GAL10, and this may be a subject of a new study, or, alternatively, if authors have any data or opinions on this issue, comments could be added to the Discussion of this paper. I recommend this paper for publication. Its length is appropriate to describe the results.

We thank the reviewer for his/her kind words and for the positive evaluation of our manuscript. We agree it would be interesting to use our method to dissect the effect of the isoforms, such as the RSC1 and RSC2 complex. Although that is outside the scope of the current study, we included a sentence to the discussion: "Moreover, our work forms a framework for future studies to understand how transcriptional bursting is regulated by the interplay of different transcriptional regulators, of different complex submodules and of complexes that vary in subunit composition or protein isoforms."

References

1. Chereji, R. V., Ocampo, J. & Clark, D. J. MNase-Sensitive Complexes in Yeast: Nucleosomes and Non-histone Barriers. *Mol. Cell* **65**, 565-577.e3 (2017).
2. Kubik, S., Bruzzone, M. J., Albert, B. & Shore, D. A Reply to "MNase-Sensitive Complexes in Yeast: Nucleosomes and Non-histone Barriers," by Chereji et al. *Mol. Cell* **65**, 578–580 (2017).
3. Brahma, S. & Henikoff, S. RSC-Associated Subnucleosomes Define MNase-Sensitive Promoters in Yeast. *Mol. Cell* **73**, 238-249.e3 (2019).
4. Floer, M. *et al.* A RSC/nucleosome complex determines chromatin architecture and facilitates activator binding. *Cell* **141**, 407–418 (2010).
5. Zhou, X., Blocker, A. W., Airoidi, E. M. & O’Shea, E. K. A computational approach to map nucleosome positions and alternative chromatin states with base pair resolution. *Elife* **5**, 1–28 (2016).

Decision Letter, first revision:

Message: Our ref: NSMB-A45827A

14th Sep 2022

Dear Dr. Lenstra,

Thank you for submitting your revised manuscript "Dynamic epistasis analysis reveals how chromatin remodeling regulates transcriptional bursting" (NSMB-A45827A). It has now been seen by the original referees and their comments are below. The reviewers find that the paper has improved in revision, and therefore we'll be happy in principle to publish it in Nature Structural & Molecular Biology, pending minor revisions to satisfy the referees' final requests and to comply with our editorial and formatting guidelines.

We are now performing detailed checks on your paper and will send you a checklist

detailing our editorial and formatting requirements in about two weeks. Please do not upload the final materials and make any revisions until you receive this additional information from us.

To facilitate our work at this stage, we would appreciate if you could send us the main text as a word file. Please make sure to copy the NSMB account (cc'ed above).

Sincerely,
Sara

Sara Osman, Ph.D.
Associate Editor
Nature Structural & Molecular Biology

Reviewer #1 (Remarks to the Author):

The authors have done a very nice job addressing my earlier comments. I appreciate the time and thought that went into their responses, and I believe the paper is now ready for publication.

Reviewer #2 (Remarks to the Author):

I am satisfied with the response and I recommend the edited manuscript for publication

Final Decision Letter:

Message 30th Mar 2023

:

Dear Dr. Lenstra,

We are now happy to accept your revised paper "Dynamic epistasis analysis reveals how chromatin remodeling regulates transcriptional bursting" for publication as a Article in Nature Structural & Molecular Biology.

Your paper will be published online soon after we receive proof corrections and will appear in print in the next available issue. You can find out your date of online publication by contacting the production team shortly after sending your proof corrections. Content is published online weekly on Mondays and Thursdays, and the embargo is set at 16:00 London time (GMT)/11:00 am US Eastern time (EST) on the day of publication. Now is the time to inform your Public Relations or Press Office about your paper, as they might be interested in promoting its publication. This will allow them time to prepare an accurate and satisfactory press release. Include your manuscript tracking number (NSMB-A45827B) and our journal name, which they will need when they contact our press office.

About one week before your paper is published online, we shall be distributing a press release to news organizations worldwide, which may very well include details of your work. We are happy for your institution or funding agency to prepare its own press release, but it must mention the embargo date and Nature Structural & Molecular Biology. If you or your Press Office have any enquiries in the meantime, please contact press@nature.com.

If you have not already done so, we strongly recommend that you upload the step-by-step protocols used in this manuscript to the Protocol Exchange. Protocol Exchange is an open online resource that allows researchers to share their detailed experimental know-how. All

uploaded protocols are made freely available, assigned DOIs for ease of citation and fully searchable through nature.com. Protocols can be linked to any publications in which they are used and will be linked to from your article. You can also establish a dedicated page to collect all your lab Protocols. By uploading your Protocols to Protocol Exchange, you are enabling researchers to more readily reproduce or adapt the methodology you use, as well as increasing the visibility of your protocols and papers. Upload your Protocols at www.nature.com/protocolexchange/. Further information can be found at www.nature.com/protocolexchange/about.

Please note that *Nature Structural & Molecular Biology* is a Transformative Journal (TJ). Authors may publish their research with us through the traditional subscription access route or make their paper immediately open access through payment of an article-processing charge (APC). Authors will not be required to make a final decision about access to their article until it has been accepted. [Find out more about Transformative Journals](https://www.springernature.com/gp/open-research/transformative-journals)

Authors may need to take specific actions to achieve [compliance with funder and institutional open access mandates](https://www.springernature.com/gp/open-research/funding/policy-compliance-faqs). If your research is supported by a funder that requires immediate open access (e.g. according to [Plan S principles](https://www.springernature.com/gp/open-research/plan-s-compliance)) then you should select the gold OA route, and we will direct you to the compliant route where possible. For authors selecting the subscription publication route, the journal's standard licensing terms will need to be accepted, including [self-archiving policies](https://www.springernature.com/gp/open-research/policies/journal-policies). Those licensing terms will supersede any other terms that the author or any third party may assert apply to any version of the manuscript.

Sincerely,
Sara

Sara Osman, Ph.D.

Associate Editor
Nature Structural & Molecular Biology
